# Metabolic reprogramming of fibro/adipogenic progenitors facilitates muscle regeneration

Alessio Reggio[1,*], Marco Rosina[1,*], Natalie Krahmer[4], Alessandro Palma[1], Lucia Lisa Petrilli[1], Giuliano Maiolatesi[1], Giorgia Massacci[1], Illari Salvatori[2], Cristiana Valle[2,3], Stefano Testa[1], Cesare Gargioli[1], Claudia Fuoco[1], Luisa Castagnoli[1], Gianni Cesareni[1,2], Francesca Sacco[1]

In Duchenne muscular dystrophy (DMD), the absence of the dystrophin protein causes a variety of poorly understood secondary effects. Notably, muscle fibers of dystrophic individuals are characterized by mitochondrial dysfunctions, as revealed by a reduced ATP production rate and by defective oxidative phosphorylation. Here, we show that in a mouse model of DMD (*mdx*), fibro/adipogenic progenitors (FAPs) are characterized by a dysfunctional mitochondrial metabolism which correlates with increased adipogenic potential. Using high-sensitivity mass spectrometry–based proteomics, we report that a short-term high-fat diet (HFD) reprograms dystrophic FAP metabolism in vivo. By combining our proteomic dataset with a literature-derived signaling network, we revealed that HFD modulates the *β*-catenin–follistatin axis. These changes are accompanied by significant amelioration of the histological phenotype in dystrophic mice. Transplantation of purified FAPs from HFD-fed mice into the muscles of dystrophic recipients demonstrates that modulation of FAP metabolism can be functional to ameliorate the dystrophic phenotype. Our study supports metabolic reprogramming of muscle interstitial progenitor cells as a novel approach to alleviate some of the adverse outcomes of DMD.

## Introduction

The interaction between the dystrophin protein and the dystrophin-associated protein complex (DAPC), which spans the sarcolemmal membrane, is essential for the integrity of the muscle fibers (Petrof et al, 1993). Dystrophin deficiency and the ensuing decrease of the DAPC components in Duchenne muscular dystrophy (DMD) patients (Emery, 1998) dramatically increases myofiber fragility upon muscle contraction and affects calcium and sodium homeostasis (Ohlendieck & Campbell, 1991; Petrof et al, 1993). Over time, chronic muscle damage culminates in the failure of the regeneration process leading to patient paralysis and, finally, death (Emery, 1998).

In addition, dystrophin deficiency also causes a variety of poorly understood secondary effects, mostly related to mitochondrial dysfunctions in myofibers. Muscles of DMD patients and animal models have a reduced oxygen consumption, spare capacity, and mitochondrial complex I activity (Percival et al, 2012; Schuh et al, 2012; Rybalka et al, 2014). Consistently, mitochondrial enzymes of the tricarboxylic acid cycle (TCA) (Lindsay et al, 2018) and of the electron transport chain (Rybalka et al, 2014) are also significantly decreased. These functional defects correlate with mitochondrial structural abnormalities. Dense and dilated mitochondria with altered *cristae* as well as swollen mitochondria have been described in muscle fibers of DMD patients (Rybalka et al, 2014).

Altogether, these and additional evidences implicate a metabolic impairment in the dystrophic disease development and progression (Rodríguez-Cruz et al, 2015). Whether a causal link between the metabolic alterations and the pathological phenotype exists remains to be established.

To counteract these metabolic alterations, different nutritional approaches have been proposed, with the aim of restoring mitochondrial functionality and muscle regeneration. A reduced caloric intake or a periodic fasting-mimicking diet were shown to stimulate regeneration of different organs, including skeletal muscle, in humans and mice (Civitarese et al, 2007; Cerletti et al, 2012; Brandhorst et al, 2015). A short-term caloric restriction enhances muscle satellite cells (MuSCs) functionality, promoting muscle regeneration upon acute muscle injury in mice (Cerletti et al, 2012). At the molecular level, the AMPK-SIRT1-PGC-1*α* axis plays a crucial role in mediating the diet-dependent increase of muscle regeneration. Consistently, pharmacological activation of AMPK by sirtuin1, resveratrol, metformin, or AICAR was shown to mitigate the dystrophic phenotype in the *mdx* mouse model of DMD (Pauly et al, 2012; Ljubicic & Jasmin, 2015; Hafner et al, 2016; Juban et al, 2018). A

[1]Department of Biology, University of Rome Tor Vergata, Rome, Italy   [2]Fondazione Santa Lucia Istituto di Ricovero e Cura a Carattere Scientifico (IRCCS), Rome, Italy   [3]Institute of Translational Pharmacology, Consiglio Nazionale delle Ricerche (CNR), Rome, Italy   [4]Department Proteomics and Signal Transduction, Max-Planck Institute of Biochemistry, Martinsried, Germany

Correspondence: francesca.sacco@uniroma2.it; cesareni@uniroma2.it; castagnoli@uniroma2.it
*Alessio Reggio and Marco Rosina contributed equally to this work

fat-enriched diet regimen was also considered as a life-style strategy to revert the metabolic impairment of DMD. Dystrophic mice fed for 16-wk with a high-fat diet (HFD) achieved an increased running ability accompanied by a reduction of myofiber necrosis without significant weight gain (Radley-Crabb et al, 2011). In addition, a variety of nutritional approaches based on amino acid supplementation have also been shown to have beneficial effects on muscle regeneration in dystrophic mouse models (Passaquin et al, 2002; Voisin et al, 2005; Barker et al, 2017; Banfi et al, 2018). Such positive effects suggest an impact of muscle metabolism and muscle homeostasis and physiology.

The skeletal muscle is a heterogeneous tissue and its regeneration after acute or chronic damage is governed by a complex interplay between muscle-resident and circulating cell populations that in concert contribute to damage resolution (Arnold et al, 2007; Christov et al, 2007; Dellavalle et al, 2011; Murphy et al, 2011).

MuSCs are the main stem progenitor cells directly responsible for the formation of new myofibers (Seale et al, 2004; Lepper et al, 2011; Sambasivan et al, 2011). However, fibro/adipogenic progenitors (FAPs), a muscle-resident interstitial stem cell population of mesenchymal origin (Vallecillo Garcia et al, 2017), are also involved in muscle regeneration (Murphy et al, 2011). FAPs play a double-edged role. In healthy conditions, they promote muscle regeneration by establishing crucial trophic interactions with MuSCs (Joe et al, 2010; Uezumi et al, 2010; Murphy et al, 2011), whereas in the late stages of the dystrophic pathology, they differentiate into fibroblasts and adipocytes. As a result, fibrotic scars and fat infiltrates compromise muscle structure and function (Uezumi et al, 2011). We considered whether any of these progenitor cell types, similarly to myofibers, have an altered metabolism that affects their function in dystrophic patients.

We have recently applied high-resolution mass spectrometry (MS)–based proteomics to characterize the changes in the FAP proteome upon acute (cardiotoxin) or chronic injury (Marinkovic et al, 2019). This unbiased strategy revealed that FAPs from *mdx* mice are also characterized by a significant reduction of mitochondrial metabolic enzymes, accompanied by an increased expression of glycolytic proteins (Marinkovic et al, 2019). Here, we demonstrate that the impaired mitochondrial metabolism of dystrophic FAPs correlates with their ability to proliferate and differentiate into adipocytes. Remarkably, in vitro metabolic reprogramming of dystrophic FAPs modulates their adipogenic potential.

As lipid-rich diets have a positive effect on the DMD phenotype, we investigated the effects of in vivo metabolic reprogramming on dystrophic FAP and MuSC biology. By applying an unbiased MS-based proteomic approach, here we show that HFD not only restores mitochondrial functionality in FAPs from dystrophic mice but also rewires key signaling networks and protein complexes. Our study reveals an unexpected connection between FAP metabolic reprogramming and their ability to promote the myogenic potential of MuSCs. The integration of our proteome-wide analysis with a literature-derived signaling network identifies β-catenin as a crucial regulator of the expression of the promyogenic factor follistatin. In summary, our study reveals that in vivo metabolic reprogramming of *mdx* FAPs correlates with a significant amelioration of the dystrophic phenotype, endorsing nutritional intervention

as a promising supportive approach in the treatment of muscular dystrophies.

## Results

### FAPs and MuSCs from dystrophic muscles have mitochondrial dysfunction and mainly rely on glycolysis to generate ATP

Recently, we have applied MS-based proteomic approach to elucidate the mechanisms underlying the different sensitivity of dystrophic FAPs to the Neurogenic locus notch homolog protein (NOTCH)-dependent adipogenesis (Marinkovic et al, 2019). Here, we dissected the proteomic dataset focusing on the expression levels of key metabolic enzymes. We found that most of the key enzymes involved in fatty acid metabolism, TCA cycle, and oxidative phosphorylation (OxPhos) are significantly down-regulated in dystrophic as compared with wild-type (*wt*) FAPs (Fig 1A and B). Conversely, many of the enzymes of the glycolytic and the pentose phosphate pathways are up-regulated (Fig 1A and B), suggesting an increased anabolic metabolism in *mdx* FAPs. To confirm and extend these observations, we collected highly pure preparations of Ly6A$^+$ (Sca1$^+$) FAPs and ITGA7$^+$ MuSCs from wild-type (*wt*) and dystrophic (*mdx*) mice, via magnetic bead cell sorting (Marinkovic et al, 2019; Reggio et al, 2019b). Sca1$^+$ FAPs and ITGA7$^+$ MuSCs express their distinctive markers PDGFRα and Pax7, respectively (Fig S1A–C).

To validate the conclusions drawn from the unbiased mass spectrometry dataset, we monitored the level of crucial metabolic enzymes. The key glycolytic and pro-anabolic enzyme pyruvate kinase M2 (PKM2) (Mazurek, 2011) is significantly increased in dystrophic FAPs (approximately fivefold) (Fig 1C and D) and to a lesser extent in dystrophic MuSCs (Fig 1E and F). In parallel, we also observed a significant reduction of mitochondrial complex V and III subunits in FAPs (Fig 1C and D) but not in MuSCs (Fig 1E and F).

To confirm that *mdx* progenitor cells have an altered energy metabolism, we purified FAPs and MuSCs and measured the oxygen consumption rate (OCR) under mitochondrial stress test conditions (Figs 2A–J and S2A–J). In comparison with *wt*, the OCR of dystrophic cells in basal conditions was reduced in *mdx* FAPs (Fig 2B). A similar, albeit smaller, decrease was observed also in *mdx* MuSCs (Fig S2B). This is in accordance with the proteomic analysis, where FAPs showed a decreased concentration of enzymes in the oxidative phosphorylation pathway (Fig 1A and B). Upon sequential injection of inhibitors of mitochondrial functions, dystrophic FAPs display a reduced mitochondrial ATP production and spare respiratory capacity, indicating that the mitochondrial efficiency and responsiveness to different energy demands are impaired in these cells (Fig 2C and D). By using the OCR/extracellular acidification rate ratio as a proxy of metabolic imbalance (Pala et al, 2018), we also conclude that *mdx* FAPs have a more robust glycolytic flux than *wt* matched cells (Fig 2E). Although less evident in comparison with *mdx* FAPs, also dystrophic MuSCs have a reduced mitochondrial performance (Fig S2C and D) and mainly exploit glycolysis for energy production (Fig S2E). Moreover, dystrophic FAPs show a lower mitochondrial trans-membrane potential and a weaker

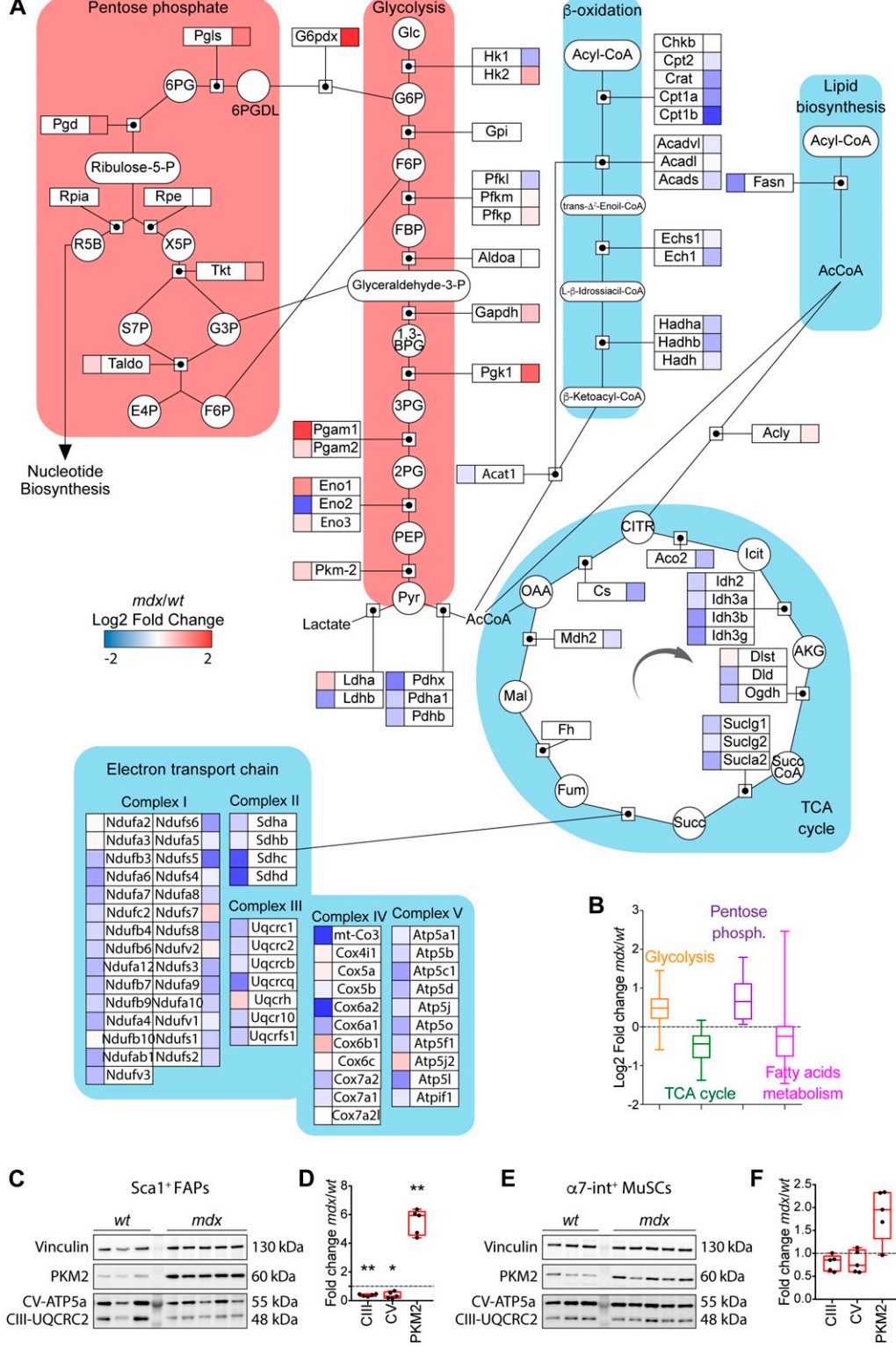

**Figure 1. Mass spectrometry–based proteomics of *mdx* fibro/adipogenic progenitors (FAPs) reveals a significant alteration of key metabolic pathways.**
**(A)** The metabolic pathway map derived from Wikipathways (http://www.wikipathways.org) of key metabolic enzymes significantly modulated in *mdx* FAPs compared with *wt*. For each detected enzyme, a corresponding square is color coded according to the log2 fold change of the protein expression level in *mdx* compared with *wt* FAPs. **(B)** Boxplot representing the log2 fold change (*mdx*/*wt*) of the abundance of metabolic enzymes annotated with the GO terms *glycolysis*, *TCA cycle*, *pentose phosphate*, and *fatty acid metabolism* in FAPs. **(C)** Western Blot of PKM2, CV-ATP5a, CIII-UQCRC2, and vinculin in FAPs isolated from the hind limbs of young (45-d old) *wt* and *mdx* mice (*wt* FAPs *n* = 3; *mdx* FAPs *n* = 5). **(D)** Bar graphs representing the fold change of the enzymes PKM2, CV-ATP5a, and CIII-UQCRC2 in FAPs. Protein levels were normalized to vinculin. **(E)** Western Blot of PKM2, CV-ATP5a, CIII-UQCRC2, and vinculin in muscle satellite cells (MuSCs) isolated from the hind limbs of *wt* and *mdx* mice (*wt* MuSCs *n* = 3; *mdx* MuSCs *n* = 5). **(F)** Bar graphs representing the fold change of the enzymes PKM2, CV-ATP5a, and CIII-UQCRC2 in MuSCs. Protein levels were normalized to vinculin. Statistical significance was estimated by *t* test. All data are represented as mean ± SEM and statistical significance is defined as *P* < 0.05; **P* < 0.01; ***P* < 0.001.

response to FCCP uncoupling treatment, as shown by the Mito-Tracker Red labelling (Fig 2K). Such reduction is not related to the change in total mitochondrial mass as shown in MitoTracker Green labelling (Fig 2L).

Our data are consistent with *wt* and dystrophic progenitor cells being in a different metabolic state. Whereas *wt* cells, similarly to other quiescent stem progenitors, mainly rely on OxPhos for energy production (Ryall et al, 2015; Knobloch et al, 2017), *mdx* progenitors

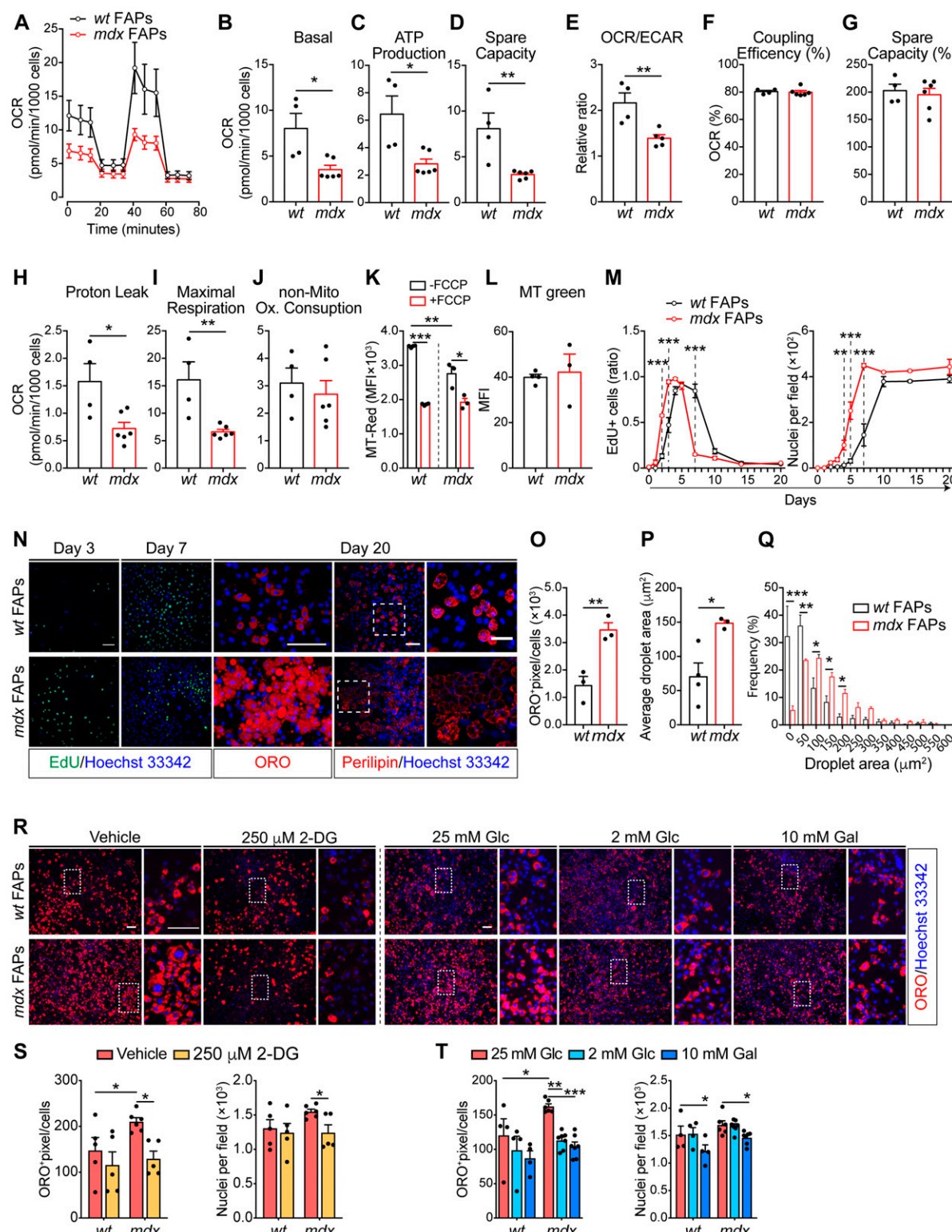

**Figure 2.  Dystrophic fibro/adipogenic progenitor (FAP) mitochondrial dysfunction correlates with an impaired ex vivo proliferation and adipogenic differentiation that can be modulated by metabolic interventions.**

**(A)** Mitochondrial stress test profile of *wt* and *mdx* FAPs (from 45-d old *wt* and mdx *mice*) by Seahorse analysis. The oxygen consumption rate (OCR) (pmol/min/$10^3$ cells) was monitored for 80 min under basal conditions and upon sequential treatment with the mitochondrial inhibitors oligomycin, FCCP, and rotenone/antimycin (*wt* FAPs $n$ = 4; *mdx* FAPs $n$ = 6). **(B, C, D, E, F, G, H, I, J)** Bar graphs representing basal OCR (B), ATP production (C), spare capacity (D), OCR/extracellular acidification rate ratio (E), coupling efficiency (% to the basal OCR) (F), spare capacity (% to the basal OCR) (G), proton leak (H), maximal respiration (I), and non-mitochondrial oxygen consumption

are less efficient in mitochondrial respiration and favor a glycolytic metabolism.

## In vitro metabolic reprogramming of dystrophic FAPs and MuSCs impact on their differentiation potential

Given the crucial role of the metabolism in controlling the stem cell fate (Knobloch et al, 2017; Pala et al, 2018), we wondered if the altered metabolic state of dystrophic muscle stem cells could impact their ability to proliferate and/or differentiate. In vitro–cultured *mdx* FAPs show an enhanced mitotic rate compared with *wt*, as revealed by the 5-ethynyl-2'-deoxyuridine (EdU) incorporation and by the growth profile (Fig 2M and N), and reduced doubling time (Fig S3A). In addition, *mdx* FAPs have a higher adipogenic differentiation potential, as revealed by the increased number of mature adipocytes (Fig 2N and O) with larger lipid droplets, 20 d after plating (Fig 2N, P, and Q).

To enquire whether the enhanced glycolytic flux in *mdx* FAPs is responsible for the increase in adipogenic propensity, we monitored *wt* and *mdx* FAP adipogenesis in conditions that restrain glycolysis. Specifically, we treated FAPs with 250 µM 2-deoxyglucose (2-DG), 2 mM glucose, or 10 mM galactose to inhibit glycolysis and shunt substrates toward OxPhos (Fig 2R). In these experimental conditions, the mitochondria functionality is improved, as shown by the enhanced ATP production (Fig S3B). Remarkably, 2-DG treatment, as well as glucose deprivation, significantly reduce the adipogenic differentiation (Fig 2R–T) and proliferation of *mdx* FAPs (Fig S3C and D), without affecting their survival (Fig S3E).

We next asked whether the in vitro metabolic reprogramming of *wt* and dystrophic MuSCs could also impact on their myogenic potential. Whereas culturing MuSCs in the presence of 2-DG does not affect myogenic differentiation (Fig S2K and L), glucose deprivation and galactose treatment significantly increase their ability to form elongated myosin heavy chain (MyHC)-positive myotubes (Fig S2K and M). Although the treatment did not impact on cell viability (Fig S2N–P), MuSC proliferation increases upon glucose reduction and galactose treatment (Fig S2Q–S) as reported by others showing that OxPhos directly influences the differentiation and proliferative capacities of MuSCs (Cerletti et al, 2012).

Overall, our data point to glycolysis as a process which plays a pivotal role in the regulation of dystrophic FAP proliferation and adipogenic differentiation. By contrast, MuSC myogenic differentiation is supported by OxPhos.

## A short-term HFD remodels FAP metabolism in *mdx* mice

In vitro, metabolic reprogramming of FAPs and MuSCs from *mdx* mice affects their differentiation potential. Given the crucial role of these progenitor cells in mediating muscle regeneration (Murphy et al, 2011), we next asked whether dietary regimens designed to activate fatty acid oxidation (FAO) and mitochondrial respiration would stimulate and restore OxPhos activity along with the pro-regenerative potential of *mdx* FAPs and MuSCs. To this aim, we fed weaned (21-d old) *mdx* and *wt* mice for 28 d with a standard HFD containing 58% kcal in fat. The control group was fed with an iso-caloric control/low-fat diet (LFD, 11% kcal in fat) (see the Materials and Methods section for details).

During the diet regimen, weight, food and water intake were recorded every 2 d (Fig S4A). Short-term treatment with the HFD was not sufficient to cause any significant changes in the body, organs, and muscle weight in either group (Fig S4B and C). Nevertheless, after 4 wk, cholesterol and triglycerides were significantly increased in both *mdx* and *wt* mice on the HFD (Fig S4D).

To elucidate the impact of short-term HFD on FAP and MuSC metabolism, we first profiled the proteome of *wt* and *mdx* FAPs and MuSCs under different diet regimens. By applying a label-free liquid chromatography (LC)-MS/MS quantitation approach (Fig 3A) (Kulak et al, 2014; Kelstrup et al, 2018), we were able to quantitate ~4,500 proteins (Fig S5A). Proteome measurements were highly accurate and reproducible with a Pearson correlation coefficient among biological replicates ranging between 0.85 and 0.95 (Fig S5B). Un-supervised hierarchical clustering (Fig S5B) and principal component analysis (Fig 3B and C) of about 4,500 proteins revealed that the proteome profiles efficiently discriminate different samples according to cell type, genetic background, and diet regimen. Interestingly, the drivers of the discrimination between *wt* and *mdx* FAPs (component 1 of the PCA loadings) were significantly enriched for proteins annotated to the cell cycle process, which is known to be up-regulated in dystrophic FAPs (Fig S5C). In addition, we also found that the drivers of the discrimination between HFD and LFD FAPs (component 2 of the PCA loadings) were significantly enriched for oxidative phosphorylation, which we expect to be up-regulated

(J) obtained by Seahorse Wave Desktop software. Statistical significance was estimated by the *t* test. **(K)** Bar graph representing the median fluorescence intensity (MFI) of MitoTracker RED (MT-Red) dye in flow cytometry in basal condition and under uncoupling with 10 µM FCCP, in *wt* and *mdx* FAPs (*n* = 3). Statistical significance was estimated by the *t* test. **(L)** Bar graph representing median fluorescence intensity (MFI) of MitoTracker GREEN (MT Green) dye in flow cytometry in basal condition on *wt* and *mdx* FAPs (*n* = 3). Statistical significance was estimated by the *t* test. **(M)** EdU labelling and growth curve profile of FAPs purified from 45-d old *wt* and *mdx* mice. FAPs were cultured for 20 d (*wt n* = 3; *mdx n* = 3). Statistical significance was estimated by two-way ANOVA. **(N)** Representative EdU (green, 20× magnification; scale bar, 100 µm), Oil Red O (ORO) staining (red, 40× magnification; scale bar, 100 µm), and confocal micrographs of perilipin immunostaining (red, 20× magnification; scale bar, 70 µm) of FAP cells at 3, 7, and 20 d. Nuclei (blue) were revealed with Hoechst 33342. **(N, O)** Bar graph resenting the adipogenic differentiation index of *wt* and *mdx* FAPs calculated as ORO-positive pixels/cell from the panel (N) (*n* = 3). **(N, P)** Bar graph representing the average lipid droplet area (µm²) of confocal images in panel (N) (*wt n* = 4; *mdx n* = 3). **(N, Q)** Bar graph representing the frequency distribution of lipid droplet areas of confocal images in panel (N) (*wt n* = 4; *mdx n* = 3). Statistical significance was estimated by *t* test. **(R)** Representative ORO staining (10× magnification; scale bar, 100 µm) of FAPs from 45-d-old *wt* and *mdx* mice. Adipogenic differentiation was obtained by incubating FAPS in adipocyte differentiation medium (ADM) followed by the adipocyte maintenance medium (AMM) in the presence of 25 mM glucose (Glc) supplemented with DMSO (vehicle) or 250 µM 2-deoxyglucose (2-DG). Alternatively, FAPs were differentiated by incubating cells with the opportune differentiation media containing either 25 mM Glc, 2 mM Glc, or 10 mM galactose (Gal). Insets are enlarged views of the dashed areas (scale bar: 100 µm). Nuclei (blue) were revealed with Hoechst 33342. **(S)** Bar plots reporting the adipogenic index (left) and the average number of nuclei per field (right) for FAPs differentiated in the presence of the vehicle or 250 µM 2-DG treatment. **(T)** Bar plots reporting the adipogenic index (left) and the average number of nuclei per field (right) for FAPs differentiated in the presence of 25 mM Glc, 2 mM Glc, or 10 mM Gal treatment (*wt n* = 3; *mdx n* = 3). Statistical significance was estimated by two-way ANOVA. All data are represented as mean ± SEM and statistical significance is defined as **P* < 0.05; ***P* < 0.01; ****P* < 0.001.

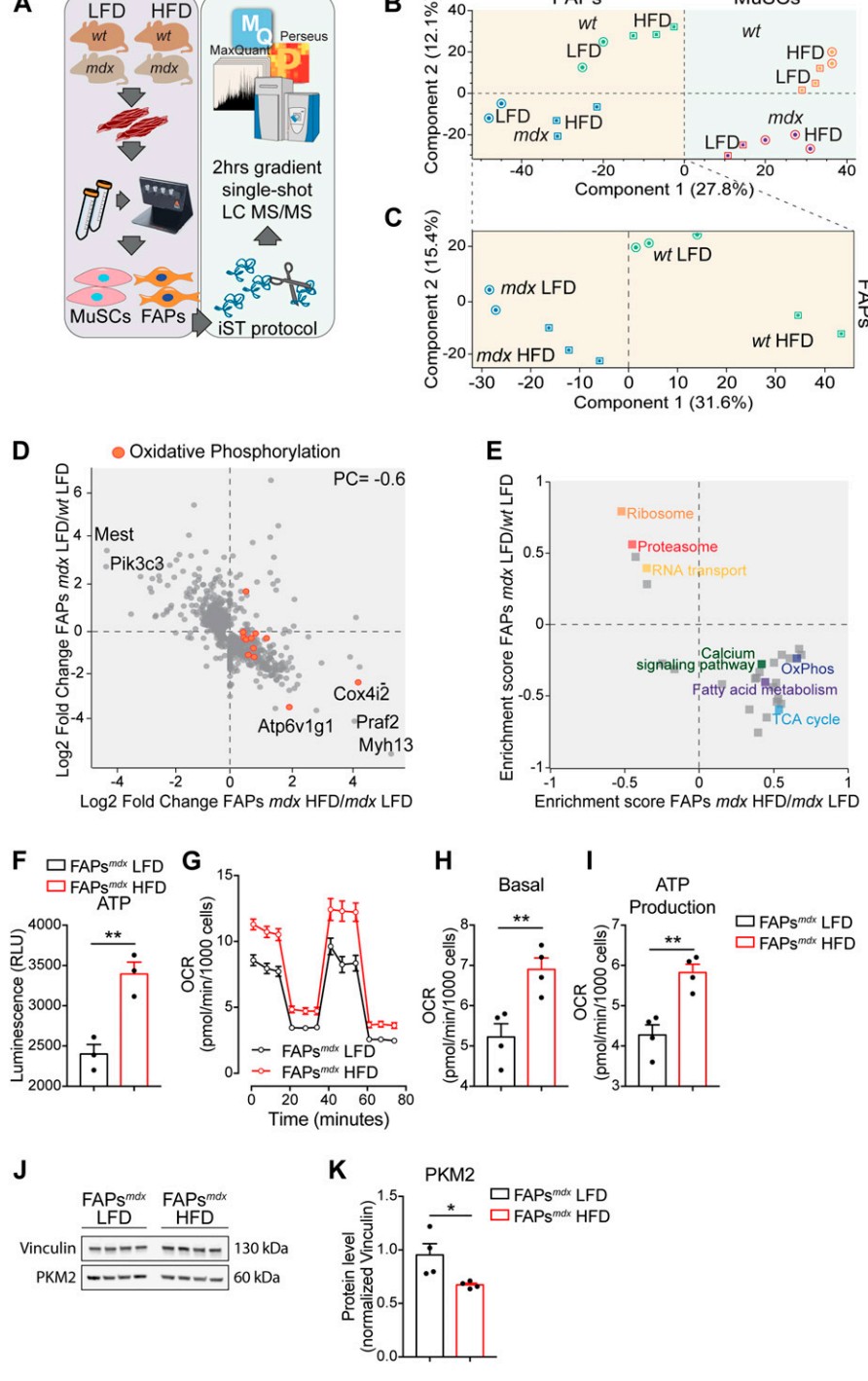

**Figure 3. Short-term high-fat diet (HFD) rewires the metabolic signature of dystrophic fibro/adipogenic progenitors (FAPs).**
**(A)** Experimental workflow to analyze the proteome of FAPs and Muscle Satellite Cells purified from 49-d-old *wt* and *mdx* mice fed with low-fat diet (LFD) and HFD (*wt* LFD *n* = 3; *wt* HFD *n* = 2; *mdx* LFD *n* = 2; *mdx* HFD *n* = 3). **(B)** Principal component analysis of the proteomic profiles of FAPs and muscle satellite cells from mice under LFD or HFD. **(C)** The principal component analysis inset shows the sample separation of *wt*/*mdx* FAPs from mice fed with LFD and HFD. **(D)** Scatterplot of the log2 fold change of protein expression level of 480 proteins significantly modulated in *mdx*/*wt* FAPs (y-axis) and HFD/LFD *mdx* FAPs (x-axis). **(E)** Two-dimensional annotation enrichment analysis of the significantly modulated proteins in *mdx*/*wt* FAPs (y-axis) and HFD/LFD *mdx* FAPs (x-axis). Groups of related GO terms are labelled with the same color, as described in the inset. **(F)** Quantitation of the ATP in FAPs purified from 49-d-old *mdx* mice fed with LFD and HFD (*n* = 3). **(G)** Mitochondrial stress test profile of LFD and HFD *mdx* FAPs. Oxygen consumption rate (pmol/min/$10^3$ cells) was monitored in real time (for 80 min) under basal condition and upon sequential treatment with mitochondrial inhibitors (*mdx* LFD *n* = 4; *mdx* HFD *n* = 4). **(H, I)** Bar graphs representing basal oxygen consumption rate (H) and ATP production (I). **(J, K)** Western blot (J) and relative densitometric analysis (K) of PKM2 and vinculin in FAPs from *mdx* mice fed with LFD and HFD (*mdx* LFD *n* = 4; *mdx* HFD *n* = 4). Statistical significance was estimated by *t* test. All data are represented as mean ± SEM and statistical significance is defined as *P < 0.05; **P < 0.01; ***P < 0.001.

in response to the HFD treatment (Fig S5C). About 9% of the proteome of both dystrophic FAPs and MuSCs was found to be significantly modulated by the diet (Fig S5D and E). The HFD significantly increased the level of proteins involved in fatty acid metabolism and OxPhos in FAPs as well as, albeit to a lesser extent, in MuSCs of dystrophic mice (Fig S5F).

We next focused on those proteins that were significantly modulated in the *mdx* model when compared with *wt*. Interestingly,

in both FAPs (Fig 3D) and MuSCs (Fig S5G), we observed an inverse correlation (PC = −0.6 in FAPs, PC = −0.7 in MuSCs) between the dystrophy-dependent proteome changes (*mdx* versus *wt*) and the diet-dependent proteome modulation (*mdx* HFD versus *mdx* LFD). This observation suggests that the HFD restores in *mdx* progenitor cells a proteome profile that is similar to the *wt* counterpart. We next used the two-dimensional annotation enrichment analyses to investigate, which biological processes were mainly restored by

HFD in *mdx* FAPs and MuSCs. Proteins involved in protein-elongation and translation, here annotated as "ribosomal," were up-regulated in *mdx* cells and reduced in concentration upon HFD (Figs 3E and S5H). In agreement with our previous observations, HFD restores the expression levels of mitochondrial proteins in both dystrophic FAPs and MuSCs, whereas FAO and OxPhos were significantly enriched only in *mdx* FAPs (Figs 3E and S5H). In agreement, most of the enzymes involved in the TCA cycle, OxPhos, and FAO were significantly up-regulated (log2 of the median fold change = 0.8) by the HFD treatment only in dystrophic FAPs and not in MuSCs (Fig S5I and J). In line with these findings, we also observed that HFD treatment significantly up-regulated PDK4, by 16-fold, in dystrophic FAPs and not in MuSCs (Fig S5K). PDK4 is a key metabolic enzyme enhancing FAO utilization through its inhibitory activity on pyruvate dehydrogenase (Palamiuc et al, 2015).

Prompted by the observed modulation of key metabolic enzymes, we asked whether the changes in the proteome profile, induced by the short-term HFD, cause a reprogramming of the metabolism in dystrophic FAPs and overcome their mitochondrial defect. To address this point, we measured the ATP levels and characterized the mitochondrial bioenergetics in cultured FAPs purified from muscles of dystrophic mice fed with HFD and LFD. Remarkably, HFD treatment increased the ATP production and improved the mitochondrial functionality of FAPs (Fig 3F–I). Consistently, the levels of PKM2 were blunted (Fig 3J and K), indicating that HFD favors the oxidative processes at the expense of glycolysis.

## Short-term HFD restores key regulatory signaling networks in *mdx* FAPs

We next asked whether, in addition to metabolism, the HFD would also affect key signaling pathways in *mdx* mice. To this end, we first selected the proteins whose abundance was affected in *mdx* and restored to values closer to *wt* by the diet. 220 and 283 proteins have such characteristics in MuSCs and FAPs, respectively (Fig S6A). Next, we mapped this subset of proteins onto a literature-derived network (Sacco et al, 2016) of signaling and physical interactions extracted from the SIGNOR (Perfetto et al, 2015) and the Mentha databases (Calderone et al, 2013). This strategy revealed key signaling networks and protein complexes whose concentrations were affected by the HFD only in dystrophic FAPs and not *wt* FAPs and in MuSCs (Figs S6B, S7A and B, and S8A and B). This network analysis showed that HFD treatment in dystrophic cells restores to *wt* level the abundance of the sarcolemmal protein, dysferlin, which plays an important role in the control of Ca²⁺-dependent sarcolemmal stability and resealing (Han et al, 2011). Our network analysis enabled us to uncover that the HFD treatment decreases the concentration levels of key proteins positively controlling FAP proliferation, which is pathologically enhanced in dystrophic muscle (Fig 4A) (Lemos et al, 2015; Marinkovic et al, 2019). Consistently, we observed a significant reduction in the number of PDGFRα-positive FAPs by labelling tibialis anterior (TA) muscle sections from dystrophic mice fed with HFD (Fig 4B). As inferred from the network modelling approach, HFD significantly suppresses the fraction of FAPs expressing the proliferation marker Ki67 (Fig 4B–D).

Interestingly, we also observed that the HFD treatment causes an up-regulation of β-catenin (Figs 4A, E, and F, and S7B), a crucial hub

controlling a variety of biological processes, including the expression level of follistatin (Jones et al, 2015) known to mediate some of the promyogenic effects of FAPs (Mozzetta et al, 2013). Consistently, two negative regulators of β-catenin, MEST/PEG1 (Li et al, 2014) and casein kinase 1α (Amit et al, 2002), are expressed at higher levels in *mdx* FAPs, whereas their modulation is reverted in FAPs from mice fed with HFD (Fig 4A, E, and F).

Next, we investigated whether the increased HFD-dependent β-catenin expression was also associated to an up-regulation of follistatin, which is significantly decreased in dystrophic FAPs in comparison with *wt* (Fig 4G). In agreement with our hypothesis, the HFD restored the mRNA of follistatin to wild-type levels (Fig 4G). In addition, we demonstrated that in vitro treatment of dystrophic FAPs with a mixture of fatty acids (50 μM of BSA-coupled palmitate, 50 μM of BSA-coupled oleate, and 100 μM carnitine) (Fig 4H) increase Ctnnb1 and Fst gene expression (Fig 4I).

To demonstrate the positive relation between β-catenin and follistatin in FAPs, we stabilized β-catenin through the high-selective GSK3 inhibitor, LY2090314 (Fig 4J) (Rizzieri et al, 2016; Kunnimalaiyaan et al, 2018). GSK3 blockade stabilizes enhances Ctnnb1 expression at 48 and 72 h while inducing a peak of Fst after 48 h of treatments (Fig 4K).

These observations encouraged us to investigate whether the HFD could increase the ability of FAPs to promote myogenic differentiation of MuSCs from dystrophic mice. Conditioned media (CM) of FAPs from mice fed with HFD are enriched in follistatin compared with LFD (Fig 4L). Independently from the diet treatment, CM from FAPs are able to promote the differentiation of MuSCs. However, media conditioned with FAPs from mice fed with the HFD better promote myogenesis of *mdx* MuSCs, when compared with media conditioned with LFD FAPs (Fig 4M–O).

Altogether, our data support a model whereby the in vivo metabolic reprogramming suppresses the FAP aberrant proliferation in dystrophic mice, while enhancing their ability to promote myogenesis of MuSCs.

## A short-term HFD ameliorates the dystrophic phenotype

Finally, we asked whether the metabolic reprogramming of muscle progenitor cells mediated by the HFD would also have an impact on the histological phenotype of dystrophic muscles. First, we monitored the serum level of creatine phosphokinase, a sensitive marker of muscle damage. Short-term HFD reduced serum creatine phosphokinase, suggesting that HFD protects *mdx* muscles from dystrophic damage (Fig 5A). Consistently, the incidence of centronuclear myofibers is reduced in dystrophic TAs and diaphragms upon HFD treatment (Fig 5B–E). Given the key role of FAPs in mediating fibrosis (Uezumi et al, 2011; Hogarth et al, 2019) in degenerating muscles, we next asked whether HFD could also ameliorate fibrosis in TAs and diaphragms. HFD decreases the abundance of fibrotic scars both in the TA and diaphragm muscles (Fig 5F–I). Intramuscular FAP-derived adipocytes are also common during the late stages of muscular dystrophy but absent in young mouse muscles (Uezumi et al, 2011; Hogarth et al, 2019). However, rare adipogenic depots are found in diaphragm muscles, independently from mouse genotypes and diet regimen (Fig S9A and B).

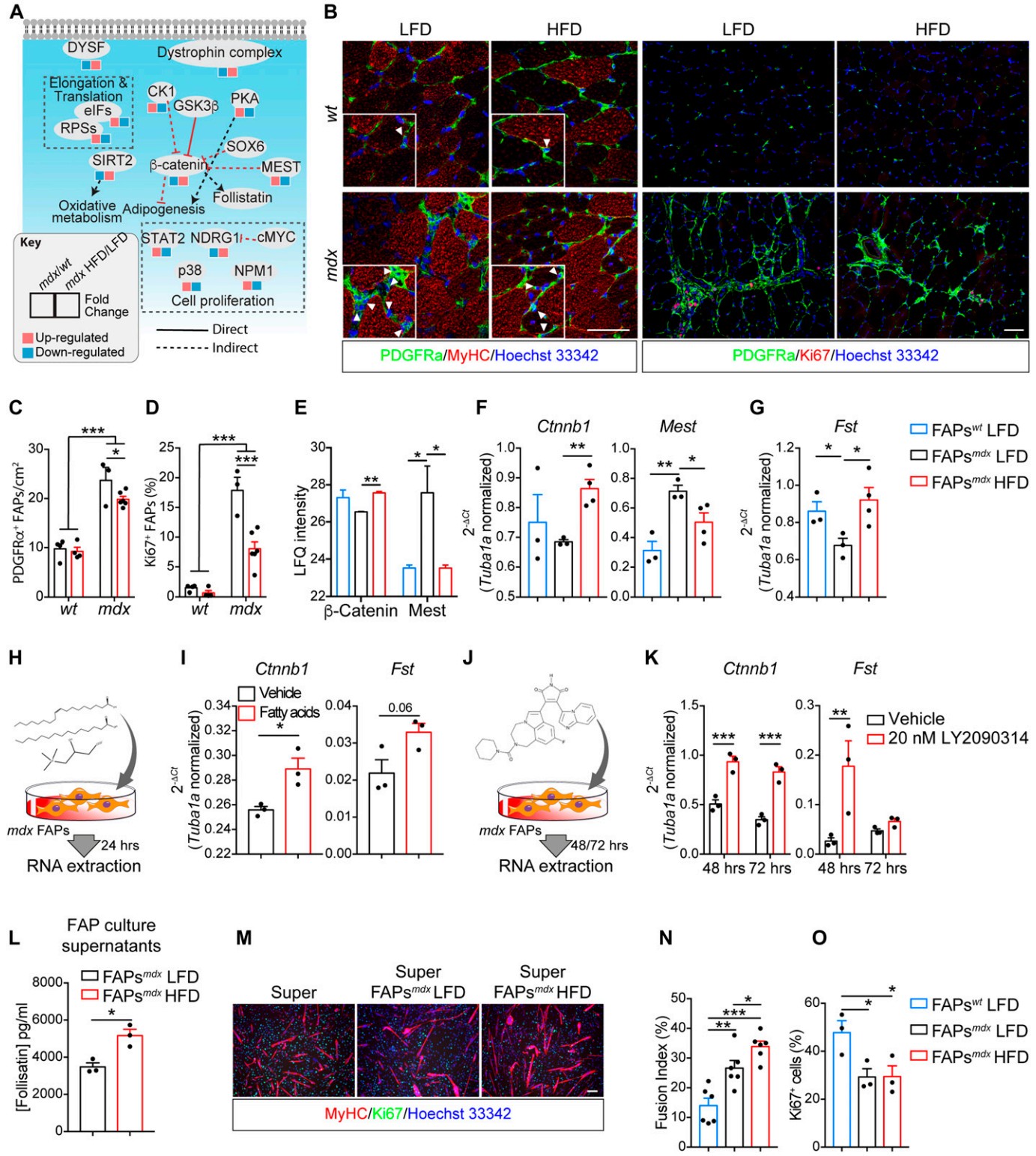

**Figure 4. Short-term high-fat diet (HFD) limits fibro/adipogenic progenitor (FAP) persistence in dystrophic muscles and restores β-catenin expression enhancing their promyogenic abilities.**

(A) Schematic representation of the main molecular events reverted by HFD treatment in *mdx* FAPs. (B) Representative confocal images of PDGFRα-positive FAPs (green) from 49-d-old *wt* and *mdx* mice fed with low-fat diet (LFD) and HFD (60× magnification; scale bar, 20 μm). Fibers (red) were stained using antibodies directed against the MyHC isoforms. Representative micrograph (20× magnification; scale bar, 100 μm) showing proliferating FAPs by coupling PDGFRα staining (green) with anti-Ki67 antibodies (red). Nuclei (blue) were revealed with Hoechst 33342. (C) Bar plot reporting the number of PDGFRα-positive FAPs per cm$^2$ of muscle section (*wt* LFD *n* = 4; *wt*

To further elucidate the role of HFD on muscle regeneration in dystrophic mice, we examined the cross-sectional area (CSA) of TAs (Fig 5J). This analysis highlighted a rescue in the *mdx* TA myofiber calibers whose average area approaches *wt* level (Fig 5K) with a significant reduction of the number of small fibers and a corresponding increase of fibers of larger calibers (Fig S9C). Although evident in TAs, such hypertrophic improvement was not appreciable in diaphragm muscles (Fig S9D–F).

The increased CSA of *mdx* TA muscles upon HFD did not result from an increased commitment and/or differentiation of MuSCs because, when cultured ex vivo, MyoD expression levels, EdU incorporation, and the fusion index do not differ, comparing MuSCs from LFD- or HFD-fed mice both at early and late culture time (Fig S9G and H). At the molecular level, we found that the increased CSA parallels an activation of the mTOR pathway, as revealed the enhanced phosphorylation of mTOR and its indirect downstream target RPS6, in whole muscle lysates (Fig 5L–N). This observation is consistent with the HFD-dependent increase of the expression level of the hypertrophic cytokine IGF1 in dystrophic FAPs (Fig 5O). Overall, we found that short-term HFD is able to ameliorate the dystrophic phenotype while triggering a hypertrophic response via the mTOR pathway.

### FAPs mediate the beneficial effects of HFD

Follistatin and IGF1 are known to mediate hypertrophy in skeletal muscle and their expression is enhanced in FAPs after short-term HFD. To ask whether the hypertrophic phenotype, observed in the *mdx* HFD muscles, can be mediated by FAPs, we transplanted *mdx* FAPs from HFD-fed *mdx* mice into the TA of syngeneic dystrophic mice. To this end, purified *mdx* FAPs from LFD- and HFD-fed mice were transduced with a lentivirus carrying a GFP-expressing vector, to allow in vivo tracing (Fig 6A). After 15 d, GFP-positive cells were found interspersed in the muscle interstitium (Fig 6B). GFP-positive cells also co-expressed the FAP distinctive marker PDGFRα, confirming that transplanted FAPs were successfully engrafted into the host muscles (Fig 6C). Remarkably, the limb receiving FAPs from HFD-fed mice displayed an increased CSA in comparison with the LFD-receiving limb (Fig 6D–G), supporting the conclusion that metabolically reprogrammed FAPs can improve the dystrophic phenotype.

These observations are consistent with a model whereby the HFD-dependent metabolic reprogramming of FAPs is responsible for the amelioration of the dystrophic phenotype that is observed in *mdx* mice fed on the HFD (Fig 7).

## Discussion

Metabolism plays a crucial role in controlling the fate of progenitor cells in tissue development, homeostasis, regeneration, and disease (Lunt & Vander Heiden, 2011; Ryall et al, 2015; Knobloch et al, 2017; Joseph et al, 2018; Pala et al, 2018; Marinkovic et al, 2019). In the skeletal muscle, a variety of nutritional approaches, including caloric restriction, fasting-mimicking drugs, long-term (16 wk) HFD, and amino acid treatments have been applied, displaying different effects on healthy and dystrophic muscle regeneration (Radley-Crabb et al, 2011; Cerletti et al, 2012; Banfi et al, 2018; Juban et al, 2018).

However, the molecular and cellular mechanisms linking metabolic reprogramming to its impact on muscle regeneration are still poorly characterized. In this context, it has been reported that the metabolic reprogramming of MuSCs tunes up the stem cells for the transition from quiescence to activation through the epigenetic control mediated by NAD-dependent deacetylase SIRT1 (Ryall et al, 2015). Moreover, mitochondrial and peroxisomal FAO are key determinants of MuSCs stem cell fate throughout the embryonal and postnatal development (Pala et al, 2018).

Here, we applied systematic approaches to investigate the role of metabolism in controlling the fate of muscle interstitial FAPs and its impact on muscle physiology and regeneration. In healthy individuals, the differentiation multipotency of FAPs is restrained by autocrine and paracrine signals (Joe et al, 2010; Uezumi et al, 2010; Lees-Shepard et al, 2018; Marinkovic et al, 2019; Reggio et al, 2019a). However, when muscle damage set regeneration in motion, FAPs release key paracrine signals targeting MuSCs and other cell partners, to trigger tissue repair (Christov et al, 2007; Joe et al, 2010; Uezumi et al, 2010; Murphy et al, 2011; Heredia et al, 2013; Lemos et al, 2015; Kuswanto et al, 2016). In pathological conditions or aging, the progressive deterioration of the mechanisms keeping FAP differentiation in check, leads to their unrestrained differentiation into fibroblasts or adipocytes, causing the formation of fibrotic scars and/or fat deposition. This outcome irreversibly alters muscle functionality (Uezumi et al, 2011; Kopinke et al, 2017; Marinkovic et al, 2019).

We have recently applied high-resolution MS-based proteomics to profile the proteome changes occurring in dystrophic FAPs isolated from *mdx* mice (Marinkovic et al, 2019). Here, we show that key metabolic enzymes involved in glycolysis are up-regulated in dystrophic FAPs, whereas the abundance of TCA and OxPhos enzymes is reduced in comparison with wild-type cells. We have further investigated the metabolic state of wild-type and dystrophic

---

HFD n = 4; *mdx* LFD n = 3; *mdx* HFD n = 6). **(D)** Bar plot reporting the fraction of Ki67-positive cells in PDGFRα-positive FAPs in TA cross-sections (*wt* LFD n = 4; *wt* HFD n = 4; *mdx* LFD n = 3; *mdx* HFD n = 6). Statistical significance was estimated by two-way ANOVA. **(E)** Mass spectrometry–based quantitation of β-catenin and Mest in *wt* and *mdx* FAPs from mice fed with LFD and HFD. **(F)** Quantitative PCR for β-catenin and Mest in *mdx* FAPs from mice fed with HFD and LFD mice. **(G)** Quantitative PCR of *Follistatin* in *wt* and *mdx* FAPs from mice fed with LFD and HFD (*wt* LFD n = 3 mice; *mdx* LFD n = 3 mice; *mdx* HFD n = 4 mice). Statistical significance was estimated by one-way ANOVA. **(H)** Representative scheme summarizing the experimental procedure to treat, ex vivo, FAPs with BSA-coupled palmitate/oleate (50 μM/50 μM) and 100 μM carnitine. **(H, I)** Quantitative PCR of Ctnnb1 and Fst transcripts in *mdx* FAPs treated as shown in (H). Statistical significance was estimated by *t* test. **(J)** Representative scheme summarizing the experimental procedure to treat, ex vivo, *mdx* FAPs with 20 nM LY2090314. **(K)** Quantitative PCR of Ctnnb1 and Fst transcripts in *mdx* FAPs treated with 20 nM LY2090314 for 48 and 72 h. Statistical significance was estimated by Two-way ANOVA (n = 3). **(L)** Bar plot reporting the concentrations of Follistatin in FAP-derived supernatants. Follistatin concentrations were analyzed via ELISA assay. **(M)** Representative immunofluorescence (20× magnification; scale bar, 100 μm) of muscle satellite cell (MuSC)–derived myotubes (red) upon incubation with the control and LFD/HFD *mdx* FAP-derived supernatants. Proliferating myoblasts (green) were detected using a Ki67 specific antibody. **(N)** Bar plot reporting the fusion index (n = 6) of differentiated MuSCs in each treatment condition. **(O)** Bar plot reporting the fraction of Ki67-positive MuSCs in each treatment condition. Statistical significance was estimated by One-way ANOVA. All data are represented as mean ± SEM and Statistical significance is defined as *P < 0.05; **P < 0.01; ***P < 0.001.

**Figure 5. Short-term high-fat diet (HFD) ameliorates the _mdx_ phenotype.**
**(A)** Serum creatine phosphokinase (units per liter, U/l) from 49-d-old _wt_ and _mdx_ mice fed with low-fat diet (LFD) or HFD (_wt_ LFD _n_ = 6; _wt_ HFD _n_ = 7; _mdx_ LFD _n_ = 6; _mdx_ HFD _n_ = 10). **(B)** Representative hematoxylin and eosin staining of TA cross-sections from 49-d-old _wt_ and _mdx_ mice fed with LFD or HFD (20× magnification; scale bar, 100 μm). **(C)** Percentage of centrally nucleated myofibers in TAs (_wt_ LFD _n_ = 8; _wt_ HFD _n_ = 6; _mdx_ LFD _n_ = 6; _mdx_ HFD _n_ = 8). **(D)** Representative hematoxylin and eosin staining of diaphragm cross-sections from 49-d-old _wt_ and _mdx_ mice fed with LFD or HFD (20× magnification; scale bar, 100 μm). **(E)** Percentage of centrally nucleated myofibers in diaphragms (_wt_ LFD _n_ = 3; _wt_ HFD _n_ = 3; _mdx_ LFD _n_ = 3; _mdx_ HFD _n_ = 3). Statistical significance was estimated by Two-way ANOVA. **(F)** Representative picrosirius red staining of TA cross-sections (20× magnification; scale bar, 100 ìm). **(G)** Bar plot showing the extent of picrosirius red area in TA cross-sections from 49-d-old _wt_ and _mdx_ mice fed

FAPs. Our analysis reveals that wild-type FAPs, similarly to other stem progenitors, mainly rely on FAO to maintain their quiescence (Ryall et al, 2015; Knobloch et al, 2017). By contrast, dystrophic FAPs, alike dystrophic fibers, have a reduced mitochondrial functionality and mainly rely on glycolysis to generate ATP (Onopiuk et al, 2009; Timpani et al, 2015).

Increasing evidence implicates modulation of glycolysis and OxPhos fluxes in the regulation of stem cell proliferation and differentiation (Ryall et al, 2015; Knobloch et al, 2017; Pala et al, 2018). Our in vitro and in vivo data support a crucial role for metabolism in the regulation of FAP biology. Modulating FAP metabolism by limiting glycolysis or enhancing mitochondrial OxPhos not only impacts cell proliferation but also corrects the aberrant activation and adipogenic differentiation of dystrophic FAPs.

Similarly to FAPs, dystrophic MuSCs show mitochondrial defects and mainly rely on glycolysis for energy production. This observation is consistent with a report showing that myoblasts derived from *mdx* mice have a reduced oxygen consumption and increased glycolysis (Onopiuk et al, 2009; Timpani et al, 2015). Interestingly, our functional and biochemical assays suggest that the metabolic differences observed between dystrophic and wild-type FAPs are more pronounced than those observed in MuSCs. In agreement with this observation, an in vivo short-term HFD treatment of dystrophic mice significantly modulates the abundance of metabolic enzymes in FAPs, although only slightly affecting MuSCs. The dramatic differences in the proteome profile of FAPs and MuSCs (more than 42% of the proteins are significantly modulated) may explain the different sensitivity of these two cell types to the diet.

Our analysis not only demonstrates an important effect of the diet on FAP metabolism but also reveals that a short-term HFD causes a significant rewire of key signaling networks.

Namely, we found that dystrophic FAPs have a reduction in the abundance of DAPC and dysferlin, similar to what has been already shown in *mdx* muscle fibers (Ohlendieck & Campbell, 1991). Interestingly, we observed that the HFD regimen restores the expression level of DAPC and dysferlin in FAPs, suggesting a correcting effect on intracellular calcium homeostasis by the activation of the surface membrane repair (Bansal et al, 2003). Because dystrophin deficiency has been already associated with myofiber mitochondrial dysfunctions (Timpani et al, 2015), the HFD-dependent recovery of the dystrophin complex in FAPs could represent one of the mechanisms contributing to the improved mitochondrial functionality. The reduced glycolytic activity of dystrophic FAPs upon HFD treatment is associated with a change in the abundance of important cell cycle regulators, including the decreased expression of p38, NPM1, STAT2, and the increase of the sirtuin SIRT2 and NDRG1, a tumor suppressor negatively regulated by cMYC (Zhang et al, 2008). Indeed, we observed

a significant reduction of PDGFRα-positive FAPs in the muscle interstitium of dystrophic mice fed with HFD as compared with the LFD counterpart. This observation suggests that HFD may attenuate FAP hyper-proliferation in dystrophic muscles.

Importantly, our network analysis revealed that the crucial hub β-catenin is also positively modulated by the diet in dystrophic FAPs. The transcriptional coactivator β-catenin plays a key role in controlling several biological processes in mesenchymal cells, including adipogenesis and fibrogenesis (Rudolf et al, 2016; Judson et al, 2018). Interestingly, HFD was already shown to regulate the expression of members of the Wnt signaling pathway in rodent models of diabetes and obesity (Chen et al, 2015). In agreement with this observation, an HFD suppresses the activity of GSK3β in an AKT-dependent manner (Wang et al, 2015). Interestingly, Anderson et al (2008) showed that GSK3β could also control mitochondrial functionality through its negative activity on PGC1a (Anderson et al, 2008). However, it is not clear yet whether the HFD treatment controls β-catenin activation via cell autonomous or non-cell autonomous mechanisms.

Noteworthy, β-catenin has been involved in the regulation of the expression of follistatin, a key myokine with promyogenic activity in MuSCs (Jones et al, 2015). Here, we demonstrate that the HFD-dependent modulation of β-catenin restores follistatin expression, which is compromised in dystrophic FAPs. Similarly, we observed that IGF1 expression was also restored by HFD in dystrophic FAPs.

IGF1 and follistatin play a key role in promoting myogenesis (Latres et al, 2005) and could stimulate myogenesis under an HFD regimen. Here, we observed the activation of the mTOR-dependent pathway that, in response to increased IGF1 production, stimulates myofiber hypertrophy. By contrast, in MuSCs isolated and differentiated ex vivo, not exposed to FAP-derived signals, the hypertrophic response is not observed. In support of this model, the supernatant of FAPs from HFD-fed *mdx* mice promotes MuSC-myotube formation more efficiently than media from control/LFD *mdx* cells. The HFD-dependent increase in MuSC myogenic activity parallels a significant amelioration of the dystrophic muscle phenotype, as highlighted by the reduced incidence of centrally nucleated myofibers, and the increase in fiber CSA. The causal link between the metabolic reprogramming and the dystrophic phenotype amelioration can be recapitulated only by injecting the HFD-derived FAPs. This observation demonstrates that FAP metabolic plasticity plays a crucial role in mediating the beneficial effects of HFD.

In agreement with this evidence, our proteomic data combined with functional metabolic analysis support a model whereby a short-term HFD reprograms the metabolism of FAPs. Upon HFD treatment, dystrophic FAPs change their metabolic state and regain their positive regulatory role in promoting muscle regeneration, by

---

with LFD or HFD (*wt* LFD $n$ = 8; *wt* HFD $n$ = 7; *mdx* LFD $n$ = 6; *mdx* HFD $n$ = 8). **(H)** Representative picrosirius red staining of diaphragm cross-sections from 49-d-old *wt* and *mdx* mice fed with LFD or HFD (20× magnification; scale bar, 100 $\mu$m). **(I)** Bar plot showing the extent of picrosirius red area in diaphragm cross-sections (*wt* LFD $n$ = 3; *wt* HFD $n$ = 3; *mdx* LFD $n$ = 3; *mdx* HFD $n$ = 3). Statistical significance was estimated by one-way ANOVA. **(J)** Representative confocal micrographs of laminin-stained (green) TA cross-sections from 49-d-old *wt* and *mdx* mice fed with LFD or HFD (20× magnification; scale bar, 70 $\mu$m), left panel. Pseudo-color representation of the myofiber caliber, ranging from 0 to 5,000 $\mu$m$^2$, right panel. Nuclei (blue) were revealed with Hoechst 33342. **(K)** Bar plot reporting the average cross-sectional area in $\mu$m$^2$ (*wt* LFD $n$ = 8; *wt* HFD $n$ = 6; *mdx* LFD $n$ = 6; *mdx* HFD $n$ = 6). Statistical significance was estimated by two-way ANOVA. **(L)** Immunoblot of TA lysates from *wt* and *mdx* mice fed with LFD and HFD. Samples were probed with anti-pmTOR (Ser2448), anti-pRPS6 (Ser240/244), anti-mTOR, anti-RPS6, and anti-vinculin antibodies. **(M)** Bar plot showing the densitometric analysis of the phosphorylation level of mTOR (posho-mTOR). **(N)** Bar plot showing the densitometric analysis of the phosphorylation level of RPS6 (posho-RPS6). Statistical significance was estimated by two-way ANOVA. **(O)** Quantitative PCR of *Igf1* in fibro/adipogenic progenitors from *wt* and *mdx* mice fed with LFD or HFD (*wt* LFD $n$ = 3; *wt* HFD $n$ = 4; *mdx* LFD $n$ = 5; *mdx* HFD $n$ = 7). Statistical significance was estimated by two-way ANOVA. All data are represented as mean ± SEM and statistical significance is defined as *$P$ < 0.05; **$P$ < 0.01; ***$P$ < 0.001.

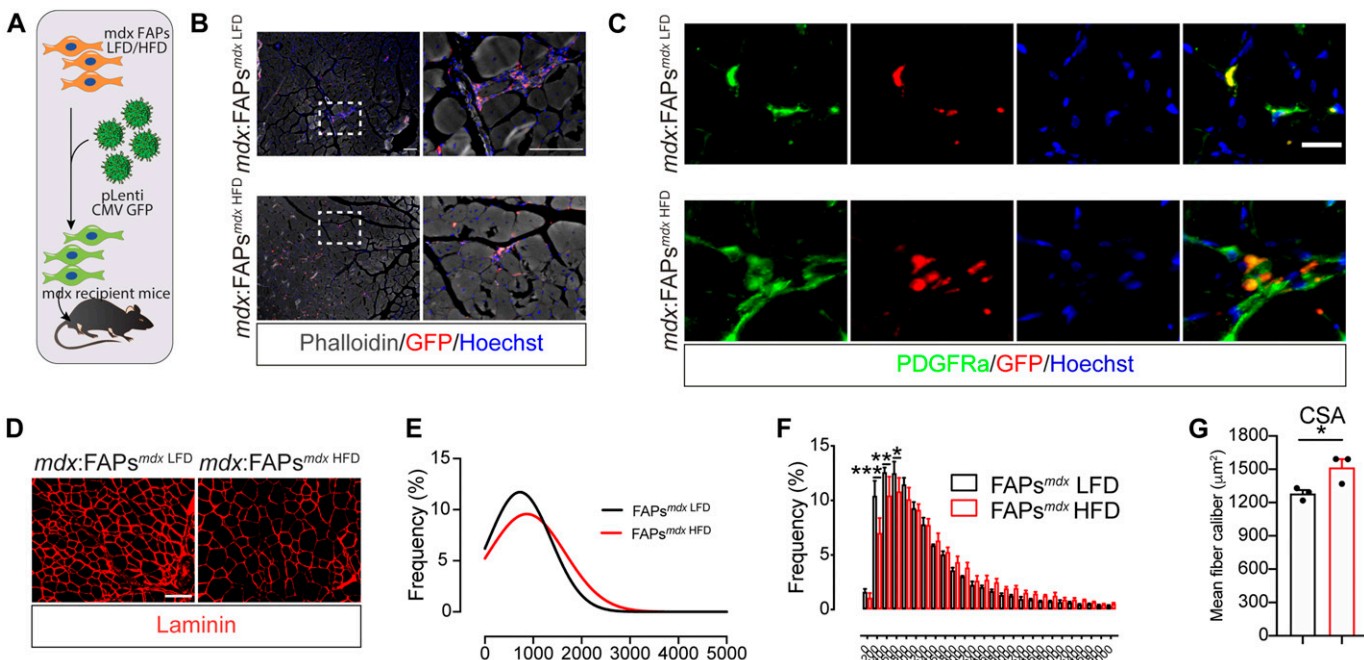

**Figure 6. Fibro/adipogenic progenitors (FAPs) mediate the beneficial effects of high-fat diet (HFD).**
**(A)** Workflow describing the FAP transduction and the transplantation strategies. FAPs were transplanted in 2-mo-old *mdx* recipient mice. FAPs (1.0 × 10⁵ cells) were resuspended in 100 μl of 1× PBS and injected in the TA of *mdx* recipient mice. **(B)** After 15 d from transplantation, GFP-positive cells (red) were found in the muscle interstitium (10× and 40× magnification; scale bar, 100 μm) (*n* = 3). Fibers (grey) were probed using phalloidin-488. **(C)** Representative immunofluorescence, showing that GFP-positive cells co-express the FAP distinctive marker PDGFRα (*n* = 3). Nuclei (blue) were revealed with Hoechst 33342. **(D)** Dystrophic (*mdx*) TA muscles receiving FAPs from LFD- and HFD-treated *mdx* mice were stained with anti-laminin antibodies to reveal the fiber outlines. **(E)** Bar plot reporting the average percentage of the frequency distribution of the myofiber areas (μm²) in each experimental condition (*n* = 3). **(E, F)** Curves, inferred on the basis of the cross-sectional area distribution in (E), showing the shift toward higher fiber calibers upon transplantation of *mdx* FAPs from mice fed with HFD (red line). **(G)** Bar plot reporting the average cross-sectional area in μm² (*n* = 3). Statistical significance was estimated by *t* test. All data are represented as mean ± SEM and statistical significance is defined as *P < 0.05; **P < 0.01; ***P < 0.001.

enhancing the MuSCs myogenic activity. Remarkably, HFD-reprogrammed FAPs are able to promote the diet-dependent amelioration of the dystrophic phenotype. In addition, ex vivo experiments suggest that metabolic reprogramming may potentially decrease FAP pathological contribution to fibrotic scar and fat infiltrate formation in the late stages of the dystrophic disease.

Recently, ApoE deficiency was shown to have catastrophic effects in dystrophic mice under HFD (Milad et al, 2017), suggesting that the local and systemic effects of dietary interventions should be carefully evaluated in preclinical animal studies. Further exploration of the proteomic resource reported here may aid in the discovery of additional mechanisms connecting metabolism to FAP biology and muscular dystrophy.

The results presented here offer a proof of principle that metabolic reprogramming of muscle progenitor cells by a HFD regimen can have a positive effect on muscle regeneration in a dystrophy mouse model. Whether these findings can be extended to dystrophy patients and whether it can be considered a viable support strategy remain to be established.

## Materials and Methods

The full list of key materials, reagents, animal strains, and software used in this work is reported in Table 1.

### Mouse models

C57BL/6J (RRID:IMSR_JAX:000664) and C57BL10ScSn-Dmd^mdx/J mice (RRID:IMSR_JAX:001801) (referred to as *wt* and *mdx*, respectively) were purchased from the Jackson Laboratories. In our study, sexes were equally balanced between genotypes. Young (45- and 49-d-old) *wt* and *mdx* mice were used in this work. Mice were bred and maintained according to the standard facility procedures. All experimental studies were conducted according to the rules of good animal experimentation I.A.C.U.C. n° 432 of March 12, 2006, and under ethical approval released on December 11, 2012, from the Italian Ministry of Health, protocol #20/01-D.

### Murine primary cells

FAPs and MuSCs were isolated from the hind limbs of male and female *wt* and *mdx* mice. Freshly sorted FAPs were resuspended in FAPs-GM consisting of DMEM GlutaMAX (25 mM Glc) supplemented with 20% FBS, 10 mM Hepes, 1 mM sodium pyruvate, and 100 U/ml penicillin/streptomycin (P/S). Freshly purified MuSCs growth medium (MuSCs-GM) consisting of DMEM GlutaMAX (25 mM Glc) supplemented with 20% FBS, 10% donor horse serum, 2% chicken embryo extract, 10 mM Hepes, 1 mM sodium pyruvate, and 100 U/ml P/S. The cells were cultivated at 37°C in 5% $CO_2$.

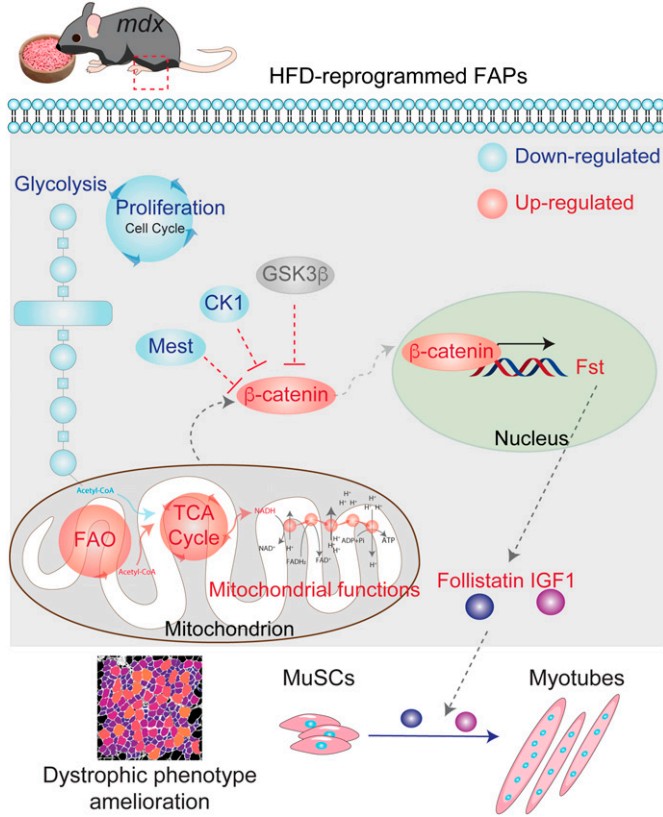

**Figure 7. Working model proposal of the beneficial effects of high-diet in the dystrophic milieu.**

A short-term high-fat diet (HFD) regimen provides a beneficial metabolic reprogramming of skeletal muscle interstitial fibro/adipogenic progenitors, dramatically affected in Duchenne muscular dystrophy. HFD restores the proper metabolic signature of dystrophic fibro/adipogenic progenitors, fueling mitochondrial pathways of fatty acid oxidation and tricarboxylic acid cycle and modulating the glycolytic flux. From the molecular point of view, $\beta$-catenin is a crucial hub that modulates muscle stem cells behavior. $\beta$-catenin inhibitors casein kinase (CK1) and MEST are repressed by HFD. The inhibition of glycogen synthase kinase 3 beta (GSK-3$\beta$) activates the $\beta$-catenin signaling in turn modulating follistatin (Fst) expression Fst, in concert with IGF1, is released to sustain the differentiation of muscle satellite cells and myotube hypertrophy. The beneficial effects of HFD lead to the amelioration of the dystrophic phenotype.

## Diet treatment

At the weaning day, *wt* and *mdx* littermate were individually housed and randomly assigned to the LFD or HFD group with free access to food and water. In both groups, sexes were equally balanced between genotypes. Mice were fed with LFD (containing 11 kcal% fat, Research Diet) and HFD (containing 58 kcal% fat, Research Diet D12331) for 4 wk (28 d). Body weight, food, and water intake were monitored every 2 d. Although LFD and HFD differed in their fat content, both diets are isocaloric. Brain, spleen, liver, adipose tissue depots, and hearth were weighted and explanted from each mouse, snap-frozen in liquid nitrogen, and stored at −80°C for further investigations.

## Isolation of MuSCs and FAPs from skeletal muscles

Hind limbs were surgically removed and then minced in HBSS (GIBCO) supplemented with 100 U/ml P/S (Roche) and 0.2% BSA (AppliChem). For each mouse, the homogeneous muscle tissue preparation was enzymatically digested in 2 $\mu$g/$\mu$l collagenase A, 2.4 U/ml dispase II, and 10 $\mu$g/ml DNase I (Roche) in Dulbecco's phosphate-buffered saline (BioWest) w/calcium and magnesium. The enzymatic digestion was performed for 1 h at 37°C with gentle shaking. The digested tissues underwent consecutive filtration through 100, 70, and 40 $\mu$m cell strainers (Corning). Before each filtration step, the cells were centrifuged at 700$g$ for 10 min at 4°C and then resuspended in fresh HBSS. Red blood cells were lysed in RBC lysis buffer (Santa Cruz). Freshly isolated muscle mono-nuclear cells were resuspended in Magnetic beads buffer (0.5% BSA and 2 mM EDTA in 1× PBS) and filtered through a 30-$\mu$m Pre-Separation Filter (Miltenyi) to remove large particles from the single cell suspension. The whole cell suspension underwent subsequential incubations with the microbead-conjugated antibodies used for the magnetic sorting. CD45$^+$ immune cells and CD31$^+$ endothelial cells were collected through the consecutive incubation with the anti-CD45 and anti-CD31 antibodies (Miltenyi). Lineage negative (Lin-) cells were incubated with anti-$\alpha$7–integrin antibodies (Miltenyi) and MuSCs selected as Lin-/$\alpha$7-int+ cells. Last, Lin-/$\alpha$7-int+ cells were incubated with anti-Sca-1 antibodies (Miltenyi) and FAPs selected as Lin-/$\alpha$7-int+/Sca-1+. The sorting procedures and the labelling procedures with the microbead-conjugated antibodies were performed according to the manufacturer's instructions.

## Lentiviral transduction and FAP transplantation

FAPs were isolated from 49-d-old *mdx* mice fed with LFD or HFD (*n* = 3) and transduced through spinoculation (Judson et al, 2018) with one MOI of pLenti CMV GFP Hygro (Addgene). Briefly, 1.0 × 10$^5$ cells were plated in serum-free medium (DMEM GlutaMax) in six-well plate and one MOI of virus was administered dropwise. The plate was centrifuged using a swinger rotor adapter for 5 min at 3,200$g$ at 25°C, then for 1 h at 2,500$g$ at 25°C. Virus-containing medium was discarded and the cells were washed twice with serum-free medium and scraped using a soft-gummy blade cell scraper. The cells were separated through centrifugation and resuspended in 100 $\mu$l 1× PBS w/Ca$^{2+}$ and Mg$^{2+}$. The cells were inoculated in TA muscles of anesthetized mice. TA muscles were collected after 15 d.

## FAP culture and differentiation

Freshly sorted FAPs were resuspended in FAPs-GM consisting of high-glucose (25 mM) DMEM GlutaMAX supplemented with 20% FBS, 10 mM Hepes, 1 mM sodium pyruvate, and 100 U/ml P/S and plated in 96-well plates at a cell density of 3.0 × 10$^4$ cells/cm$^2$. After 3/4 d, the FAP-GM was fully refreshed and cells cultured for two additional days before the induction of the adipogenic differentiation. The adipogenic differentiation was induced incubating FAPs with the adipocyte differentiation medium (ADM: FAPs-GM supplemented with 1 $\mu$g/ml insulin, 0.5 mM 3-isobutyl-1-methylxanthine, and 1 $\mu$M dexamethasone) for 3 d followed by two additional days in adipocyte maintenance medium (AMM: FAP-GM supplemented with 1 $\mu$g/ml insulin). Unstimulated cells were maintained in fresh FAP-GM. For allowing spontaneous FAP growth and adipogenesis, the cells were plated in 96-well plates at a density of 7.5 × 10$^3$ cells/cm$^2$

**Table 1. List of key materials, reagents, animal strains, and software.**

| Reagent or resource | Source | Identifier |
|---|---|---|
| Antibodies | | |
| Mouse mAb anti-vinculin (SPM227) | Abcam | Cat. no. ab18058; RRID:AB_444215 |
| Total OXPHOS Rodent WB antibody cocktail | Abcam | Cat. no. ab110413; RRID:AB_2629281 |
| Goat polyAb anti-mouse IgG (H+L)-HRP | Bio-Rad | Cat. no. 1721011; RRID:AB_11125936 |
| Goat polyAb anti-rabbit IgG (H+L)-HRP | Bio-Rad | Cat. no. 1706515; RRID:AB_11125142 |
| Rabbit mAb Ki67 (D3B5) | Cell Signaling Technology (CST) | Cat. no. 9129; RRID:AB_2687446 |
| Rabbit mAb anti-non-phospho active $\beta$-catenin (Ser33/37/Thr41) (D13A1) | CST | Cat. no. 8814; RRID:AB_11127203 |
| Rabbit mAb anti-PKM2 (D78A4) XP | CST | Cat. no. 4053; RRID:AB_1904096 |
| Rabbit mAb anti-PPARγ (81B8) | CST | Cat. no. 2243; RRID:AB_823598 |
| Rabbit polyAb anti-perilipin1 (D418) | CST | Cat. no. 3470; RRID:AB_2167268 |
| MYH1E antibody (MF 20) | Developmental Studies Hybridoma Bank | Cat. no. MF 20; RRID:AB_2147781 |
| PAX7 antibody | Developmental Studies Hybridoma Bank | Cat. no. PAX7; RRID:AB_2299243 |
| F(ab')2-goat anti-mouse IgG (H+L) cross-adsorbed secondary antibody, Alexa Fluor 555 | Invitrogen | Cat. no. A-21425; RRID:AB_2535846 |
| Goat anti-mouse IgG (H+L) cross-adsorbed secondary antibody, Alexa Fluor 488 | Invitrogen | Cat. no. A-11001; RRID:AB_2534069 |
| Alexa Fluor 488 phalloidin | Invitrogen | Cat. no. A12379 |
| a7-Integrin MicroBeads, mouse | Miltenyi Biotec | Cat. no. 130-104-261 |
| CD31 MicroBeads, mouse | Miltenyi Biotec | Cat. no. 130-097-418 |
| CD45 MicroBeads, mouse | Miltenyi Biotec | Cat. no. 130-052-301 |
| Sca1 MicroBeads, mouse | Miltenyi Biotec | Cat. no. 130-106-641 |
| Goat polyAb anti-PDGFRa | R&D Systems | Cat. no. AF1062; RRID:AB_2236897 |
| Rabbit polyAb MyoD (M-318) | Santa Cruz Biotechnology | Cat. no. sc-760; RRID:AB_2148870 |
| Rabbit polyAb GFP antibody (FL) | Santa Cruz Biotechnology | Cat. no. sc-8334; RRID:AB_641123 |
| Rabbit polyAb anti-Lamin | Sigma-Aldrich | Cat. no. L9393; RRID:AB_477163 |
| Donkey anti-goat IgG(H+L), multi-species SP ads-Alexa Fluor 488 | SouthernBiotech | Cat. no. 6425-30 |
| Goat anti-mouse IgG(H+L), human ads-Alexa Fluor 555 | SouthernBiotech | Cat. no. 1031-32 |
| Goat anti-rabbit IgG(H+L), mouse/human ads-Alexa Fluor 488 | SouthernBiotech | Cat. no. 4050-30 |
| Goat F(ab')2 anti-mouse Ig, human ads-Alexa Fluor 647 | SouthernBiotech | Cat. no. 1012-31 |
| CD140a (PDGFRA) monoclonal antibody (APA5), APC | Thermo Fisher Scientific | Cat. no. 17-1401-81; RRID:AB_529482 |
| ITGA7 monoclonal antibody (334908), APC | Thermo Fisher Scientific | Cat. no. MA5-23555; RRID:AB_2607368 |
| Pax7 antibody (SPM613) (FITC) | Novus Biologicals | Cat. no. NBP2-47923F |
| Chemicals, peptides and recombinant proteins | | |
| pLenti CMV GFP Hygro (656-4) | Addgene | Cat. no. 17446-LVC |
| Seahorse XF Base Medium | Agilent Technologies | Cat. no. 102353-100 |
| Seahorse XF96 Flux Pack | Agilent Technologies | Cat. no. 102416-100 |
| Bovin Serum Albumin (BSA) Fraction V | AppliChem | Cat. no. A1391 |
| EDTA | Applichem | Cat. no. A5097 |
| Killik, embedding medium for criostate blue | Bio-Optica | Cat. no. 059801 |
| Clarity Western ECL Blotting Substrates | Bio-Rad | Cat. no. 10070-5061 |
| Criterion TGX Gradient gel 4–15% | Bio-Rad | Cat. no. 5671085 |

**Table 1.  Continued**

| | | |
|---|---|---|
| Criterion TGX Gradient gel 4–20% | Bio-Rad | Cat. no. 5671095 |
| Mini-PROTEAN TGX gel 4–15% | Bio-Rad | Cat. no. 4561086 |
| Mini-PROTEAN TGX gel 4–20% | Bio-Rad | Cat. no. 4561096 |
| Trans-Blot Turbo Midi Nitrocellulose Membrane | Bio-Rad | Cat. no. 1704157 |
| Trans-Blot Turbo Mini Nitrocellulose Membrane | Bio-Rad | Cat. no. 1704156 |
| DPBS 1× w/Ca$^{2+}$, Mg$^{2+}$ | BioWest | Cat. no. L0625-500 |
| DPBS 1× w/o Ca$^{2+}$, Mg$^{2+}$ | BioWest | Cat. no. L0615-500 |
| Cell-Tak Cell and Tissue adhesive | Corning | Cat. no. 354240 |
| Falcon 100-$\mu$m cell strainer | Corning | Cat. no. 352360 |
| Falcon 40-$\mu$m cell strainer | Corning | Cat. no. 352340 |
| Falcon 70-$\mu$m cell strainer | Corning | Cat. no. 352350 |
| Matrigel basement membrane mix | Corning | Cat. no. 356234 |
| Eukitt mounting medium | Electron Microscopy Sciences | Cat. no. 15320 |
| Donor horse serum | Euroclone | Cat. no. ECS0090D |
| Fetal bovin serum (FBS) | Euroclone | Cat. no. ECS0180L |
| Goat serum | Euroclone | Cat. no. ECS0200D |
| Non-fat dried milk | Euroclone | Cat. no. EMR180500 |
| Trypsin–EDTA 0.05% w/phenol red | Euroclone | Cat. no. ECM0920D |
| Dulbecco's Modified Eagle's Medium (DMEM) GlutaMAX | GIBCO | Cat. no. 61965-026 |
| Dulbecco's Modified Eagle's Medium (DMEM) no glucose, no glutamine, phenol red | GIBCO | Cat. no. A1443001 |
| GlutaMAX Supplement | GIBCO | Cat. no. 35050038 |
| HBSS w/o Ca$^{2+}$, Mg$^{2+}$ | GIBCO | Cat. no. 14170112 |
| Penicillin–streptomycin (P/S) 10,000 U/ml | GIBCO | Cat. no. 15140122 |
| RPMI 1640 | GIBCO | Cat. no. 21875-091 |
| Cell culture microplate, black | Greiner Bio One | Cat. no. 655090 |
| Hoecsth 33342 | Invitrogen | Cat. no. H3570 |
| MitoTracker Green FM | Invitrogen | Cat. no. M7514 |
| MitoTracker Red CMXRos | Invitrogen | Cat. no. M7512 |
| Novex NuPAGE Sample Reducing Agent (10×) | Invitrogen | Cat. no. NP0009 |
| NuPAGE LDS Sample Buffer (4×) | Invitrogen | Cat. no. NP0007 |
| TRIzol | Invitrogen | Cat. no. 15596026 |
| Sodium fluoride (NaF) | Millipore | Cat. no. 67414 |
| MS columns | Miltenyi Biotec | Cat. no. 130-042-201 |
| Pre-separation filters (30 $\mu$m) | Miltenyi Biotec | Cat. no. 130-041-407 |
| Histo-Clear Solution | National diagnostics | Cat. no. HS-200 |
| Ortovanadate | New England Biolabs | Cat. no. P0758S |
| 11 kcal% fat w/sucrose Surwit Diet (LFD) | Research diet | |
| 58 kcal% fat w/sucrose Surwit Diet (HFD) | Research diet | |
| Cyto-Grow | Resnova | Cat. no. TGM-9001-A |
| Collagenase A | Roche | Cat. no. 11088793001 |
| Dispase II | Roche | Cat. no. 04942078001 |
| DNase I | Roche | Cat. no. 11284932001 |
| Paraformaldehyde solution (PFA) 4% in PBS | Santa Cruz | Cat. no. sc-281692 |

| | | |
|---|---|---|
| RBC Lysis buffer 10× | Santa Cruz Biotechnology | Cat. no. sc-296258 |
| LY2090314 | Selleckchem | Cat. no. S7063 |
| Chicken embryo extract | Seralab | Cat. no. CE-650-J |
| 2-deoxyglucose (2-DG) | Sigma-Aldrich | Cat. no. D6134 |
| 3-Isobutyl-1-methylxanthine (IBMX) | Sigma-Aldrich | Cat. no. I5879 |
| Antimycin | Sigma-Aldrich | Cat. no. A8674 |
| D-Galactose | Sigma-Aldrich | Cat. no. G0750 |
| D-Glucose | Sigma-Aldrich | Cat. no. G8270 |
| Dexamethasone | Sigma-Aldrich | Cat. no. D4902 |
| Dimethyl sulfoxide (DMSO) Hybri-Max | Sigma-Aldrich | Cat. no. D2650 |
| EGTA | Sigma-Aldrich | Cat. no. E3889 |
| Eosin | Sigma-Aldrich | Cat. no. E4009 |
| FCCP | Sigma-Aldrich | Cat. no. C2920 |
| Glutamine 200 mM | Sigma-Aldrich | Cat. no. G7513 |
| Hematoxylin | Sigma-Aldrich | Cat. no. H3136 |
| Hepes 1 M | Sigma-Aldrich | Cat. no. H0887 |
| Inhibitor phosphatase mixture II | Sigma-Aldrich | Cat. no. P5726 |
| Inhibitor phosphatase mixture III | Sigma-Aldrich | Cat. no. P0044 |
| Insulin solution human | Sigma-Aldrich | Cat. no. I9278 |
| Magnesium chloride | Sigma-Aldrich | Cat. no. M8266 |
| Oil Red O | Sigma-Aldrich | Cat. no. O0625 |
| Oleic acid | Sigma-Aldrich | Cat. no. O1008 |
| Oligomycin | Sigma-Aldrich | Cat. no. O4876 |
| Palmitic acid | Sigma-Aldrich | Cat. no. P0500 |
| Phenylmethanesulfonyl fluoride (PMSF) | Sigma-Aldrich | Cat. no. P7626 |
| Protease inhibitor mixture | Sigma-Aldrich | Cat. no. P8340 |
| Rotenone | Sigma-Aldrich | Cat. no. R8875 |
| Sodium chloride | Sigma-Aldrich | Cat. no. S5150 |
| Sodium dodecyl sulfate | Sigma-Aldrich | Cat. no. L3771 |
| Sodium pyruvate 100 mM | Sigma-Aldrich | Cat. no. S8636 |
| TERGITOL Solution NP-40 | Sigma-Aldrich | Cat. no. NP40S |
| Triton X-100 | Sigma-Aldrich | Cat. no. T9284 |
| Trizma base | Sigma-Aldrich | Cat. no. T1503 |
| Tween 20 | Sigma-Aldrich | Cat. no. P1379 |
| Critical commercial assays | | |
| Click-iT EdU Alexa Fluor 488 Imaging Kit | Invitrogen | Cat. no. C10337 |
| CellTiter-Glo luminescent cell viability assay | Promega | Cat. no. G7571 |
| Custom mouse follistatin ELISA | RayBiotech | Cat. no. ELM-FST-1 |
| In situ cell death detection lit, TMR red | Sigma-Aldrich | Cat. no. 12156792910 |
| Deposited data | | |
| Proteome of *wt* and *mdx* FAPs from mice fed with LFD and HFD | This paper | N/A |
| Proteome of *wt* and *mdx* MuSCs from mice fed with LFD and HFD | This paper | N/A |

**Table 1.  Continued**

| Proteome of *wt* and *mdx* FAPs | Marinkovic et al (2019) | N/A |
|---|---|---|
| Experimental models: organisms/strains | | |
| Mouse C57BL/6J | Jaxmice | Cat. no. JAX:000664; RRID:IMSR_JAX: 000664 |
| Mouse C57BL10ScSn-Dmdmdx/J | Jaxmice | Cat. no. JAX:001801; RRID:IMSR_JAX: 001801 |
| Software and algorithms | | |
| Illustrator CC 2017 | Adobe; http://www.adobe.com/products/illustrator.html | RRID:SCR_010279 |
| CytExpert software | Beckman Coulter | RRID:SCR_017217 |
| Photoshop CC 2017 | Adobe; https://www.adobe.com/products/photoshop.html | RRID:SCR_014199 |
| GraphPad Prism | GraphPad Software; http://www.graphpad.com/ | RRID:SCR_002798 |
| Cytoscape 3.7.1 | http://cytoscape.org; Shannon et al (2003) | RRID:SCR_003032 |
| Fiji | http://fiji.sc; Schindelin et al (2012) | RRID:SCR_002285 |
| MaxQuant 1.5.3.6 | http://www.biochem.mpg.de/5111795/maxquant; Cox and Mann (2008) | RRID:SCR_014485 |
| Perseus 1.6.2.3 | http://www.perseus-framework.org; Tyanova et al (2016) | RRID:SCR_015753 |
| Seahorse Wave Desktop software | Agilent; http://www.agilent.com/en-us/products/cell-analysis-(seahorse)/software-download-for-wave-desktop | RRID:SCR_014526 |

in FAPs-GM. The medium was refreshed every 2 d for 20 d. Metabolic perturbations were performed during the differentiation period by supplementing ADM and AMM with 250 $\mu$M of 2-DG or by replacing Glu with Gal. Palmitate and oleate were dissolved in ethanol at 200 mM. Each fatty acid was diluted separately at 3 mM in a solution of 10% BSA in PBS 1× w/Ca$^{2+}$, Mg$^{2+}$, and incubated overnight at 37°C in gentle shaking. Solution were filtered through a syringe filter of 0.22 $\mu$m. Separate stock solution were aliquoted and stored at −20°C. Freshly thawed aliquots were used for treatments. Pharmacological blockade of GSK3$\beta$ was performed at the onset of the adipogenic differentiation, by supplementing ADM with 20 nM LY2090314. Further experiments involving the use of LY2090314 were performed using in vitro subcultured FAPs. In this case, FAPs were grown in Cyto-Grow (Resnova) by plating 2.5 × 10$^3$ cells/cm$^2$. The cells were not passaged more than once and routinely checked for their multipotency. Near-to-confluent cells were dethatched and cultured as described above in FAP-GM.

## MuSC culture and differentiation

To reduce fibroblasts contamination, freshly sorted *wt* and *mdx* MuSCs were pre-plated for 2 h in pre-warmed MuSC Growth Medium (MuSCs-GM) consisting of high-glucose (25 mM) DMEM GlutaMAX supplemented with 20% FBS, 10% donor horse serum, 2% chicken embryo extract, 10 mM Hepes, 1 mM sodium pyruvate, and 100 U/ml P/S. Pre-plated MuSCs were seeded in MuSC-GM at a cell density of 1.5 × 10$^4$ cells/cm$^2$ in Matrigel-coated 96-well plates. After 3 d, MuSC-GM was fully refreshed and cells cultured for three additional days allowing spontaneous myotube formation. Metabolic perturbations

were carried out in the last 3 d of MuSC differentiation by incubating cells with the MuSC-GM supplemented with 250 $\mu$M of 2-DG or by replacing Glu with Gal.

## Follistatin (FST) ELISA assay

The Follistatin concentration in FAP-derived culture supernatants was assessed using Custom Mouse Follistatin ELISA Kit (RayBiotech) according to the manufacturer's instructions. FST standard curve was prepared through serial dilution. FAP CM were diluted 1:4 with Assay Diluent. 100 $\mu$l of standard/samples were loaded in duplicate. Colorimetric assay was performed at 450 nm.

## FAP-derived CM

Freshly sorted FAPs from mice fed with LFD and HFD were plated in FAPs-GM at a density of 3.0 × 10$^4$ cells/cm$^2$. After 4 d of culture, CM were recovered, centrifuged at 3,000$g$ to remove cells, and stored at −80°C before use. The control CM was prepared by incubating empty wells with the same medium. CM treatments were performed on 2-d cultured *mdx* MuSCs. Briefly, MuSC-GM was removed and replaced by FAP-derived CMs. Cells were fixed 3 d later.

## Proteome sample preparation

Cells were lysed in sodium deoxycholate (SDC) lysis buffer containing 4% (wt/vol) SDC, 100 mM Tris–HCl (pH 8.5). Proteome preparation was done using the in StageTip (iST) method (Kulak et al, 2014). Samples

were separated by HPLC in a single run (without pre-fractionations) and analyzed by MS.

## LC-MS/MS measurements

The peptides were separated on a reverse-phase column (50 cm, packed in-house with 1.9-$\mu$m C18-Reprosil-AQ Pur reversed-phase beads) (Dr Maisch GmbH) over 120 min (single-run proteome analysis). Eluting peptides were electrosprayed and analyzed by tandem MS on a Q Exactive HF (Thermo Fischer Scientific) using higher-energy collisional dissociation (HCD)-based fragmentation, which was set to alternate between a full scan followed by up to five fragmentation scans.

## Proteome data processing

Raw MS data were analyzed in the MaxQuant environment (Cox & Mann, 2008), version 1.5.1.6, using the Andromeda engine for database search. MS/MS spectra were matched against the Mus Musculus UniProt FASTA database (September 2014), with an false discovery rate (FDR) of <1% at the level of proteins, peptides, and modifications. Enzyme specificity was set to trypsin, allowing cleavage of N-terminal to proline and between aspartic acid and proline. The search included cysteine carbamidomethylation as a fixed modification, and N-terminal protein acetylation and oxidation of methionine. Label-free proteome analysis was performed in MaxQuant. Where possible, the identity of peptides present but not sequenced in a given run was obtained by transferring identifications across LC-MS runs ("match between runs"). Up to three missed cleavages were allowed for protease digestion and peptides had to be fully tryptic.

## Proteome bioinformatics data analysis

Bioinformatics analysis was performed in the Perseus software environment (Tyanova et al, 2016). Statistical analysis of proteome was performed on logarithmized intensities for those values that were found to be quantified in any experimental condition. To identify significantly modulated proteins, missing values (proteins not identified or not quantified) were imputed by using a normal distribution. Then, we performed a $t$ test with a $P$-value cutoff of 0.05 and S0 = 0.1. Categorical annotation was added in Perseus in the form of gene ontology (GO) biological process (GOBP), molecular function (GOMF), and cellular component (GOCC), and KEGG pathways and Keywords (extracted from Uniprot). We performed a 2D annotation enrichment analyses to identify statistically significant GO-terms, KEGG pathways, and Keywords enriched in *mdx* HFD FAPs (Cox & Mann, 2008). Multiple hypothesis testing is controlled by using a Benjamini-Hochberg FDR threshold of 0.07. Then for each term, the corresponding $P$-value and score are assigned. Whereas a score near 1 indicates a positive enrichment, a score near –1 means a negative enrichment of the category. The complete proteome measurements for FAPs and MuSCs are collected in Tables S1 and S2, respectively.

## Network analysis

This strategy has been previously developed and applied by our group (Sacco et al, 2012, 2016). Kinase–substrate relationships and physical interactions were extracted from SIGNOR and MINT, respectively (Zanzoni et al, 2002; Perfetto et al, 2015), and were mapped onto the complete human proteome in Cytoscape (Shannon et al, 2003). Then the network was first filtered to maintain only relationships between proteins that were significantly modulated in *mdx* LFD FAPs compared with *wt* LFD FAPs and in *mdx* HFD FAPs compared with *mdx* LFD FAPs.

## Flow cytometry

Immunophenotyping of Sca1+ FAPs a was performed by staining 4.0 × $10^5$ cells with anti-CD140a (APC) (PDGFR$\alpha$) antibody for 30 min at 4°C. Immunophenotyping of ITGA7+ MuSCs was performed by staining 4.0 × $10^5$ cells with anti-ITGA7 (APC) antibody for 30 min at 4°C. Subsequently, MuSCs were permeabilized with 0.3% Triton-X 100 in 1× PBS for 15 min at 4°C. Permeabilized cells were incubated with anti-Pax7 (FITC) for 30 min at 4°C. The cells were washed twice and analyzed using CytoFlex Cytometer (Beckman Coulter). Data were analyzed using the Cytexpert Software (Beckman Coulter).

Freshly purified *wt* and *mdx* FAPs were resuspended in FAPs-GM and incubated in the absence/presence of 10 $\mu$M carbonyl cyanide-p-trifluoromethoxyphenylhydrazone (FCCP) for 30 min at 37°C. After mitochondrial uncoupling, FAPs were incubated with 200 nM of MitoTracker Red CMXRos or MitoTracker Green FM for 20 min. FAPs were washed twice with 1× PBS and fixed in 0.5% PFA before flow cytometry analysis. For each sample, 10,000$^3$ events were recorded using BD FACSCALIBUR (BD Biosciences).

## ATP measurement

Freshly isolated FAPs were seeded at a cell density of 6 × $10^4$ cells/cm$^2$ in 96-well plates. After 20 h, ATP content was evaluated with CellTiter-Glo Luminescent Viability Assay (Promega) according to the manufacturer's instructions. Briefly, CellTiter-Glo reagent was added at a ratio of 1:1 to cell culture medium and mixed for 2 min on an orbital shaker. Samples were transferred to a black-walled 96-well plate and luminescence was recorded through Victor Multilabel plate reader (PerkinElmer). Background consisted of FAP-GM incubated in the absence of FAPs and processed as described above. Data are presented as mean relative luminescence units.

## Bioenergetics analysis of FAPs and MuSCs

Freshly isolated *wt* and *mdx* FAPs/MuSCs were plated on Seahorse XFe96 Microplates (Agilent Technologies) at the density of 1.4 × $10^5$ cells/cm$^2$ and cellular bioenergetics assessed 4 d later. Alternatively, subcultured FAPs were plated at the density of 7.0 × $10^4$. After 24 h, the cells were incubated in Cyto-Grow:RPMI in 1:1 ratio for 20 h to stimulate mitochondrial functionality before assessing the cellular bioenergetics. Cartridges were hydrated overnight with Seahorse XF Calibrant and incubated at 37°C in the absence of $CO_2$. Calibrant was changed and refreshed 1 h before the assay. Mitochondrial stress test was performed according to Agilent's recommendations. Briefly, the cells were washed four times with Seahorse XF Base Medium supplemented with 10 mM glucose, 1 mM sodium pyruvate, and 2 mM glutamine (pH 7.4 ± 0.01). The cells were incubated at 37°C

in the absence of $CO_2$ for 1 h. Mitochondrial inhibitors were sequentially injected at the following final concentrations: 1 $\mu$M oligomycin, 1.5 $\mu$M FCCP, and 1 $\mu$M/1 $\mu$M Rotenone/Antimycin (Sigma-Aldrich).

## Immunoblot analysis

FAPs were washed in PBS 1× and stored at −80°C or directly lysed in ice-cold radio-immunoprecipitation assay lysis buffer (150 mM NaCl, 50 mM Tris–HCl, pH 7.5, 1% Nonidet P-40, 1 mM EGTA, 5 mM MgCl₂, and 0.1% SDS) supplemented with 1 mM PMSF, 1 mM ortovanadate, 1 mM NaF, protease inhibitor mixture 1×, inhibitor phosphatase mixture II 1×, and inhibitor phosphatase mixture III 1×. Protein lysates were separated at 15,500$g$ for 30 min. The total protein concentration was determined using the Bradford reagent. Protein extracts were denatured and heated at 95°C for 10 min in NuPAGE LDS Sample Buffer that contained DTT as a reducing agent (NuPAGE Sample Reducing Agent). According to the needs, proteins were resolved using either 4–15% or 4–20% Bio-Rad Mini-PROTEAN TGX/CRITERION polyacrylamide gels. Proteins were transferred to Trans-Blot Turbo Mini or Midi Nitrocellulose Membranes using a Trans-Blot Turbo Transfer System (Bio-Rad), and the nonspecific binding membranes were saturated for 1 h at RT in blocking solution (5% skimmed milk powder, 0.1% Tween 20 in 1× TBS). Saturated membranes were incubated overnight with the specific primary antibodies diluted in blocking solution. Host-specific HRP-conjugated secondary antibodies were diluted in blocking solution and used for the detection of the primary antibodies. Chemiluminescent detection was performed using Clarity Western ECL Blotting Substrates (Bio-Rad) and the Las-3000 Imaging System (Fujifilm). Band densities were quantified using Fiji and normalized to the loading control. All antibodies were diluted according to the manufacturer's instructions.

## Real-time PCR

Total RNA was extracted using TRIzol from TAs, diaphragms, and FAPs. Muscles were minced in short pieces and homogenized with a Dounce tissue grinder directly in TRIzol. Before resuspension, total RNA was precipitated overnight in the presence of 10 $\mu$g of glycogen. RNA concentration was assessed using NanoDrop Lite Spectrophotometer. Total RNA (ranging between 300 and 1,000 ng) was reverse-transcribed into cDNA with PrimeScript RT Reagent Kit. qPCR reactions were carried out with SYBR Premix Ex Taq (Tli RNaseH Plus) and performed in technical duplicates for each biological repeat. Each reaction mixture (final volume of 20 $\mu$l) contained 50 ng of cDNA. Relative expression was estimated by $2^{-\Delta Ct}$ as described (Livak & Schmittgen, 2001). To compare multiple runs, mRNA levels were determined by the $2^{-\Delta\Delta Ct}$ method (Livak & Schmittgen, 2001). *Actb* and *Tuba1a* were used as reference genes. The full primer list is reported in Table 2.

## Histological analysis

TA and diaphragm muscles were embedded in optimal cutting temperature medium and snap-frozen in liquid nitrogen. Embedded muscles were stored at −80°C before cryosectioning.

Prior optimal cutting temperature embedding, for GFP immunolabelling in transplanted TAs, muscles were fixed in 4% PFA for 4 h at RT and incubated for 4 h in sucrose gradient solutions (10%, 20% and 30%). Muscles were transverse cryosectioned at 10 $\mu$m using a Leica cryostat. Muscle sections were sequentially collected in different glass slides, thus adjacent sections were separated at least 100 $\mu$m from each other. Cryosections were performed for the entire length of the muscles.

For the hematoxylin and eosin stain (H&E), sections were fixed in 4% PFA for 10 min at RT. Fixed sections were washed twice in 1× PBS and incubated in hematoxylin solution for 20 min. Sections were rinsed for 10 min in tap water and washed in ultrapure water. Samples were incubated with the eosin solution for 1 h. Finally, stained sections were ethanol-dehydrated, clarified with the Histo-Clear solution and mounted with the Eukitt mounting medium.

## Immunofluorescence

Cells and sections were fixed in 2% PFA for 15 min at RT. For immunocytochemistry fixed cells were washed twice in 1× PBS and incubated in permeabilization solution (0.5% Triton X-100 in 1× PBS) for 5 min. Unspecific binding were saturated by incubating samples in blocking solution (10% FBS and 0.1% Triton X-100 in 1× PBS) for 1 h. The cells were incubated overnight at 4°C in gentle shaking in the presence of the primary antibodies. The cells were washed four times with 0.1% Triton X-100 in 1× PBS and incubated, for 30 min at RT, with host-specific secondary antibodies. Primary and secondary antibodies were diluted according to the manufacturer's recommendations in blocking solution. Finally, the cells were washed four times with 0.1% Triton X-100 in 1× PBS and counterstained for 5 min at RT with the Hoechst solution (1 mg/ml Hoechst 33342 and 0.1% Triton X-100 in 1× PBS). Cells were washed twice with 1× PBS and stored at 4°C.

For muscle section immunofluorescence, different protocols were adapted according to the antigen of interest. For laminin staining, fixed sections were washed twice in 1× PBS and permeabilized (0.3% Triton X-100 in 1× PBS) for 30 min at RT. Permeabilized slices were incubated in blocking solution (10% goat serum, 1% glycine, and 0.1% Triton X-100 in 1× PBS) for 1 h at RT. Blocked specimen were incubated overnight at 4°C in the presence of anti-laminin (Sigma-Aldrich) diluted 1:200 in blocking solution.

For the in situ detection of PDGFR$\alpha$, fixed and permeabilized sections were blocked for 30 min in protein block serum-free reagent (DAKO) and then incubated overnight with anti-PDGFR$\alpha$ diluted 1:80 in a carrier solution (4% BSA in 1× PBS). After incubation with the primary antibodies, the sections were washed four times with 0.1% Triton X-100 in 1× PBS and incubated for 30 min at RT with host-specific secondary antibodies. Nuclei were revealed by incubating samples for 10 min with Hoechst solution. The sections were washed twice in 1× PBS and mounted with Aqua Poly/Mount mounting medium.

## TUNEL assay

In situ detection of apoptotic nuclei was performed through the In Situ Cell Death Detection Kit, TMR red (Sigma-Aldrich). Cells were fixed with 2% PFA for 20 min at RT and permeabilized with 0.5% Triton X-100 in 1× PBS for 5 min at RT. Positive controls were treated with DNAse I (3 U/ml in 50 mM Tris–HCl, pH 7.5, and 1 mg/ml BSA) for

**Table 2. List of primers used in real time PCR experiments.**

| Gene | ID | Forward primer | Reverse primer |
|---|---|---|---|
| Primer list | | | |
| *Actb* | 11461 | 5′-CACACCCGCCACCAGTTCGC-3′ | 5′-TTGCACATGCCGGAGCCGTT-3′ |
| *C/ebpa* | 12606 | 5′-GAGGGGAGGGACTTAGGTGT-3′ | 5′-GGAGGTGCAAAAAGCAAGGG-3′ |
| *Ctnnb1* | 12387 | 5′-TGGACCCTATGATGGAGCATG-3′ | 5′-GGTCAGTATCAAACCAGGCCAG-3′ |
| *Igf1* | 16000 | 5′-GCTGGTGGATGCTCTTCAGT-3′ | 5′-TCCGGAAGCAACACTCATCC-3′ |
| *Mest* | 17294 | 5′-GGCCATTGGATCCTATAAATCCGTA-3′ | 5′-GGTAGTGGCTAATGTGGTCATCCAG-3′ |
| *Myh3* | 12883 | 5′-TCGTCTCGCTTTGGCAA-3′ | 5′-TGGTCGTAATCAGCAGCA-3′ |
| *Pparg*1 | 19016 | 5′-CGAGTGTGACGACAAGGTGA-3′ | 5′-ACCGCTTCTTTCAAATCTTGTCTG-3′ |
| *Pparg*2 | 19016 | 5′-GCCTATGAGCACTTCACAAGAAAT-3′ | 5′-GGAATGCGAGTGGTCTTCCA-3′ |
| *Tuba1a* | 22142 | 5′-AAGCAGCAACCATGCGTGA-3′ | 5′-CCTCCCCCAATGGTCTTGTC-3′ |

10 min at RT. Negative controls were treated only with the enzyme buffer without transferase enzyme. All wells were incubated with 50 μl of reaction solution for 1 h at 37°C in humidified chamber. After labelling, the samples were washed with 1× PBS and counterstained with Hoechst 33342. Images were acquired as described in "Image acquisition."

### EdU incorporation assay

For EdU labelling (Invitrogen), the cells were seeded at a concentration of $6.0 \times 10^3$ cells/cm$^2$ and incubated in the presence of 10 μM of 5-ethylnyl-2′-deoxyuridine (EdU) the day prior fixation. Click-iT reaction was performed according to the manufacturer's instructions.

### Oil Red O (ORO)

Oil Red O (Sigma-Aldrich) stock solution was prepared according to the manufacturer's recommendations. Fixed and permeabilized cells were washed twice with 1× PBS and incubated for 10 min with filtered ORO working solution (3:2 ratio ORO:ultrapure water). Stained cells were washed twice for 10 min with 1× PBS and counterstained using Hoechst 33342. ORO-stained cells were acquired via fluorescence microscopy.

### Image acquisition

Immunolabeled cells and section were acquired using the DMI6000B fluorescent microscope (Leica). Cell Micrographs were captured in unbiased fashion with the "matrix screener" mode, by acquiring nine non-overlapping field across the well.

Section micrographs were captured with DMI6000B and Nikon Eclipse TE300 microscopes (Nikon). At least three independent fields were captured in five nonadjacent sections for each mouse. Acquisition were performed at 40×, 20×, or 10× magnification.

Representative micrographs were captured using the confocal microscope Olympus IX-81 at 20× magnitude.

For bright-field microscopy, at least three non-overlapping micrographs of five independent H&E–stained sections were captured using a Zeiss Lab A1 AX10 microscope at the 20× magnification.

### Biochemical analysis

Before sacrifice, a blood sample was collected from each mouse in SST microtainers by retro-orbital sampling. For biochemical clinical analysis, blood samples were incubated for 20 min at RT and sera separated at 15,500*g* for 7 min. Non-fasting glucose, triglycerides, cholesterol, creatine kinase, and creatinine were measured using the automatic analyzer Keylab (BPC BioSed Srl).

### Histological analysis

Centrally nucleated myofibers were counted and scored manually using Fiji (Schindelin et al, 2012) by three independent collaborators in blind analysis. Results are expressed as ratio over the total myofibers.

CSA measurements of myofibers were conducted on laminin-stained sections. The myofiber diameters were evaluated using a Fiji plug-in and data represented as average CSA and frequency distribution, for each mouse. For the pseudo-color representation of the CSA, images were processed with the ROI Color Coder macro using Fiji (http://imagejdocu.tudor.lu/doku.php?id=macro:rol_color_coder) (Kopinke et al, 2017). The manual analysis of the images was performed in blind.

### Seahorse data processing

OCR was normalized by estimating the number of cells for each experimental sample. Briefly, immediately after the assay, the cells were fixed with 2% PFA for 15 min. Nuclei were labelled with Hoechst 33342 and a representative field of each well was captured and automatically scored with CellProfiler (Carpenter et al, 2006). An estimation of the total nuclei number for each condition was calculated, correlating the field to the well areas. OCR values are expressed as pmol $O_2$/min/$10^3$ cells. Mitochondria activity parameters were extracted with Agilent Wave Software (version 2.6.0).

### Cell differentiation measurements

Adipogenic differentiation of FAPs was estimated in unbiased fashion using CellProfiler through a dedicated pipeline that

recognize, for each captured field, the ORO-positive area (expressed in pixels). The modules of the pipeline were routinely adjusted in agreement with the image magnifications. The average ORO-positive area was normalized over the average number of cells per field. Alternatively, the percentage of adipogenic differentiation was evaluated by counting the number of adipocytes over the total cells in each field. Positive objects were manually scored using Fiji by two independent collaborators.

Adipocyte dimension was evaluated by measuring the area of each lipid droplet associated with each adipocyte. Briefly, three randomly and non-overlapping field of perilipin-stained adipocytes were acquired at 40× magnification. The area of each lipid droplet was delimited using the "Frehand selection" tool of Fiji. At least 15 adipocytes for each field were included in the analysis.

Myogenic differentiation of MuSCs was scored manually using Fiji, by calculating the fusion index. Fusion index was defined as the ratio between the number of nuclei included into MyHC-expressing cells (containing at least three nuclei) over the number of nuclei for each field.

The percentage of EdU-positive cells was estimated in unbiased fashion through a dedicated pipeline using CellProfiler.

Before automated analysis, a supervised control of each Cell-Profiler module was performed to ensure the proper recognition of the markers of interest. The manual analysis of the images was performed in blind.

## Statistical analysis

Data are represented as mean ± SEM of at least three independent experimental samples unless otherwise mentioned. In addition, cell studies were further performed at least in technical duplicates. Statistical significance between two groups was estimated using the unpaired $t$ test assuming a two-tailed distribution. Multiple comparisons between three or more groups were performed using one-way or two-way ANOVA. Statistical significance is defined as *$P$ < 0.05; **$P$ < 0.01; ***$P$ < 0.001. All statistical analyses were performed using Prism 7 (GraphPad).

## Contact for reagent and resource sharing

Further information for reagents may be directed to, and will be fulfilled by the lead contact, Francesca Sacco (francesca.sacco@uniroma2.it).

## Data and software availability

The datasets generated and/or analyzed in the current study are available from the corresponding authors on reasonable request.

# Supplementary Information

# Acknowledgements

This work was supported by a grant of the European Research Council (grant number 322749 [G Cesareni]) and by Rita Levi Montalcini grant, Ministero dell'Istruzione, Università e Ricerca (MIUR) (F Sacco). This work was partly supported by a grant from the Italian association for cancer research (AIRC) to G Cesareni (Investigator Grant - IG 2013). A Reggio was supported by Fondazione Umberto Veronesi. The authors warmly thank Prof Maria Rosa Ciriolo, from the Department of Biology of the University of Rome Tor Vergata for her kind contribution during the critical evaluation of the manuscript.

## Author Contributions

A Reggio: conceptualization, data curation, formal analysis, investigation, visualization, methodology, and writing—original draft, review, and editing.
M Rosina: conceptualization, data curation, formal analysis, investigation, visualization, methodology, and writing—original draft, review, and editing.
N Krahmer: investigation and methodology.
A Palma: investigation and methodology.
LL Petrilli: investigation.
G Maiolatesi: investigation.
G Massacci: investigation.
I Salvatori: investigation and methodology.
C Valle: resources, investigation, methodology, and writing—review and editing.
S Testa: resources, investigation, methodology, and writing—review and editing.
C Gargioli: investigation, visualization, and methodology.
C Fuoco: investigation, visualization, and methodology.
L Castagnoli: conceptualization, resources, supervision, investigation, visualization, methodology, and writing—review and editing.
G Cesareni: conceptualization, resources, supervision, funding acquisition, visualization, methodology, project administration, and writing—review and editing.
F Sacco: conceptualization, resources, data curation, formal analysis, supervision, funding acquisition, visualization, methodology, project administration, and writing—original draft, review, and editing.

## Conflict of Interest Statement

The authors declare that they have no conflict of interest.

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
