## [Reviewer comments · Life Science Alliance]

Metabolic reprogramming of Fibro/Adipogenic Progenitors facilitates muscle regeneration

Alessio Reggio, Marco Rosina, Natalie Krahmer, Alessandro Palma, Lucia Lisa Petrilli, Giuliano Maiolatesi, Giorgia Massacci, Illari Salvatori, Cristiana Valle, Stefano Testa, Cesare Gargioli, Claudia Fuoco, Luisa Castagnoli, Gianni Cesareni, and Francesca Sacco

DOI: <https://doi.org/10.26508/lsa.202000660>

Corresponding author(s): Francesca Sacco, Max Planck Institute and Gianni Cesareni, University of Rome Tor Vergata

Review timeline:

Submission Date:	2020-01-13
Editorial Decision:	2020-01-20
Revision Received:	2020-01-27
Accepted:	2020-01-28

Scientific Editor: Andrea Leibfried

Transaction Report:

Please note that the manuscript was previously reviewed at another journal and the reports were taken into account in the decision-making process at Life Science Alliance. Since the original reviews are not subject to Life Science Alliance's transparent review process policy, the reports and author response cannot be published.

RE: Life Science Alliance Manuscript #LSA-2020-00646P

Prof. Gianni Cesareni
University of Rome Tor Vergata
Department of Biology
Via della Ricerca Scientifica
Rome, italia 133
Italy

Dear Dr. Cesareni,

Thank you for submitting your revised manuscript entitled "Metabolic reprogramming of Fibro/Adipogenic Progenitors facilitates muscle regeneration". Your manuscript was reviewed at another journal twice before, and you provided us with the reviewer reports and your response to the concerns raised.

In this manuscript, you build upon your previous work and show that fibro/adipogenic progenitors of mdx mice have a different metabolic profile as compared to fibro/adipogenic progenitors in wild-type mice. You also show that short-term treatment of mdx mice with a high fat diet affects the proliferative potential of the fibro/adipogenic progenitors, ameliorating the muscle phenotype in this Duchenne mouse model.

The reviewers who evaluated your work thought that the present manuscript is rich in data and solid, but they would have expected data on a causal link between muscle phenotype rescue and the observed altered beta-catenin signaling/altered follistatin expression. We appreciate that extensive resources are needed to do so via mouse genetics and that you also already tried to affect follistatin action with blocking antibodies. The commercial anti-follistatin antibodies did not work in this assay, unfortunately.

We think that the lack of a causal link and the remaining reviewer concerns do not preclude publication in Life Science Alliance, and we would thus like to invite you to submit a final version of your manuscript to us. The remaining reviewer concerns should get addressed via text changes as you propose. In addition:

- Please make sure to list all authors within our submission system when uploading the final version and to have all corresponding authors link their ORCID iD to their profiles in our system
- Please make sure to fill in all mandatory fields in our submission system- these are highlighted with an asterisk
- Please provide the supplementary tables mentioned in the manuscript
- Please re-name the "key resources table" to either Table 1 or as an Supplementary table
- Please include the legends to the supplementary figures and the suppl. references in the main manuscript docx file
- The magnification in Fig 2R Vehicle / mdx FAPs does not match the origin box, which should be moved slightly further down.
- Please add scale bars to Fig 4B,M, Fig 5 B,D,F,G,J,K, S3C
- Please add a legend for figure 7

A. FINAL FILES:

B. MANUSCRIPT ORGANIZATION AND FORMATTING:

Thank you for your attention to these final processing requirements. Please revise and

format the manuscript and upload materials within 7 days.

Sincerely,

RE: Life Science Alliance Manuscript #LSA-2020-00646-TRR

Dr. Francesca Sacco
Max Planck Institute
Proteomics and Signal Transduction
Am Klopferspitz 18
Martinsried, Munich D82152
Germany

Dear Dr. Sacco,

Thank you for submitting your Research Article entitled "Metabolic reprogramming of Fibro/Adipogenic Progenitors facilitates muscle regeneration". I appreciate the introduced changes and it is a pleasure to let you know that your manuscript is now accepted for publication in Life Science Alliance. Congratulations on this interesting work.

DISTRIBUTION OF MATERIALS:

Again, congratulations on a very nice paper. I hope you found the review process to be constructive and are pleased with how the manuscript was handled editorially. We look forward to future exciting submissions from your lab.

Sincerely,

Andrea Leibfried, PhD
Executive Editor
Life Science Alliance

Meyerhofstr. 1
69117 Heidelberg, Germany
t +49 6221 8891 502
e a.leibfried@life-science-alliance.org
www.life-science-alliance.org